# Immunotherapy that improves response to chemotherapy in high-grade serous ovarian cancer

Samar Elorbany [1] ✉, Chiara Berlato[1], Larissa S. Carnevalli[2], Eleni Maniati [1], Simon T. Barry [2], Jun Wang[1], Ranjit Manchanda [3,4], Julia Kzhyshkowska[5,6] & Frances Balkwill [1]

Single-cell RNA sequencing (scRNAseq) of tumour-infiltrating immune cells in high-grade serous ovarian cancer (HGSOC) omental biopsies reveals potential targets that could enhance response to neo-adjuvant chemotherapy (NACT). Analysis of 64,097 cells identifies NACT-induced overexpression of stabilin-1 (clever-1) on macrophages and FOXP3 in Tregs that is confirmed at the protein level. STAB1 inhibition in vitro induces anti-tumour macrophages. FOXP3 anti-sense oligonucleotide (FOXP3-ASO), repolarises Tregs to an effector T cell phenotype. ScRNAseq on 69,781 cells from an HGSOC syngeneic mouse model recapitulates the patients' data. Combining chemotherapy with anti-stabilin1 antibody and/or Foxp3-ASO significantly increases survival of mice with established peritoneal disease in two HGSOC syngeneic models and progression-free survival in a third model. Long-term survivors (300 days + ) are resistant to tumour rechallenge. Anti-stabilin1 antibody enriches the tumours with CXCL9+ macrophages and Foxp3-ASO increases TBET cell infiltration. Our results suggest that targeting these molecules in immune cells may improve chemotherapy response in patients.

High grade serous ovarian cancer (HGSOC) is the commonest type of ovarian cancer, characterised by high somatic copy number alterations[1] and intra-tumour heterogeneity[2]. Patients with homologous DNA repair deficient (HRD) tumours have good responses to platinum-based therapy and benefit from PARP inhibitors but 50% of the patients are not in this category. HGSOC tumours with high fold back inversion (FBI) are characterised by extensive epithelial to mesenchymal transition (EMT), naïve T cell infiltration and low tumour antigenicity[3]. These patients represent a particular treatment challenge due to their poor response to platinum-based chemotherapy.

We, and others, have reported that neoadjuvant chemotherapy (NACT) has an impact on immune cells in HGSOC. NACT has potential immune-stimulating effects on T cells[2,4] and B cells[5]. We previously reported a decline in tumour-associated macrophages (TAMs), defined as CD3-/CD19-/CD14 + /HLADR+ cells, CD206 and CD163 post-NACT in HGSOC patients and mouse models. However, CSFR1 inhibitors reduced progression-free survival in mice[6].

Single-cell RNA sequencing (scRNA-seq) revealed the complexity of myeloid cells in different human tumours beyond the dichotomous anti-tumour/pro-tumour classification as most subpopulations express features of both[7,8]. In view of this complexity, more information about the effect of chemotherapy on tumour-associated macrophages is needed to identify and prioritise new myeloid targets.

[1]Barts Cancer Institute, Queen Mary University of London, Charterhouse Square, London, UK. [2]Bioscience, Early Oncology, AstraZeneca, Cambridge, UK. [3]Wolfson Institute of Population Health, Queen Mary University of London, London, UK. [4]Department of Gynaecological Oncology, Barts Health NHS Trust, London, UK. [5]Institute of Transfusion Medicine and Immunology, Mannheim Institute for Innate Immunosciences (MI3), Medical Faculty Mannheim, Heidelberg University, Mannheim, Germany. [6]German Red Cross Blood Service Baden-Württemberg-Hessen, Mannheim, Germany. ✉ e-mail: s.elorbany@qmul.ac.uk

Most studies on immune cells in HGSOC focus on CD8 cells[2,9,10] with less information on the role of CD4 cells, especially after chemotherapy. Regulatory T cells (Tregs) represent 10–50% of the CD4 cells[11] in several human tumours. Their significance is controversial, some studies linking their presence to poor prognosis e.g., in ovarian[12], melanoma and gastric cancers[13], and others to good prognosis e.g., in colorectal cancer[14,15].

Using site-matched, patient unmatched HGSOC metastases from seven patients, we analysed 64,097 immune enriched cells to study the effect of platinum-based NACT on the immune cells in the omental tumour microenvironment (TME). We used this information to identify targets that could potentially enhance the effects of chemotherapy and validated these targets in a larger cohort of matched pre- and post-treatment patient samples. Furthermore, we analysed 69,781 cells from control and chemotherapy-treated omental tumours from a syngeneic HGSOC mouse model that replicated the human TME[16]. We then validated the efficacy of our targets in combination with chemotherapy in three mouse models of HGSOC.

## Results

To study the potential effect of chemotherapy on HGSOC, we performed scRNA-seq on immune enriched cells from omental metastasis from seven patients; three patients had primary debulking surgery (PDS) (28,962 cells) and four patients received NACT (35,135 cells) (Supplementary Data 1).

### NACT has pro- and anti-tumour effects on macrophages

After integration and filtering, 14,672 single cells were annotated as monocytes/macrophages in eleven clusters (Fig. 1A). Clusters were named after their top upregulated genes (Fig. 1B, Supplementary Data 2). Cluster M0 had highest expression of *STAB1*, *SIGLEC1* and *ISG15* and cluster M1 had high expression of *SELENOP* and *GPNMB* genes. Macrophages in cluster M2 expressed MHCII and complement genes, whereas Clusters 4 and 5 additionally expressed T cell chemoattractant genes *CCL3/CCL4* and *CXCL9/10/11*, respectively. Clusters M3 and M7 were characterized by *S100A8/9* expression and other monocyte/early macrophage genes. Clusters M8 and M9 showed high expression of *MMP9* and M6 represented proliferating macrophages. Finally, M10 expressed high levels of metallothionine genes. Individual patients had variable percentages of each macrophage cluster with expansion of clusters M0, M1, M2, M4 after NACT (Supplementary Fig. S1A, B).

We used geneset variation analysis (GSVA) to study the differentially enriched pathways in the macrophage clusters (Fig. 1C). Macrophage clusters split into two main categories. Clusters M2-M5 were significantly enriched in antigen-presentation, T/B cell activation and complement pathways (APC clusters). The remaining clusters were enriched in ECM-regulation and angiogenesis pathways (ECM-regulating clusters). Comparing APC clusters with ECM-regulating clusters we found significant downregulation of phagosome-specific proteases and phagosome acidification genes, e.g., *CTSB*, *CTSD*, *CTSL* and *TCIRG1*, which are responsible for target antigen degradation (Supplementary Fig. S1C, D and Supplementary Data 3) with upregulation of inflammatory and T cell chemoattractant genes. This suggested that the ECM-regulating macrophage clusters degrade the phagocytosed antigen with reduced ability for antigen presentation.

Macrophage trajectory analysis using Slingshot[17] showed a continuum starting at monocytes and early macrophages clusters (M3 and M7) and maturing towards either APC macrophage clusters or ECM-regulatory clusters (Supplementary Fig. S1E).

Compared to PDS samples, we found significant upregulation of macrophage scavenger receptors, e.g., *STAB1*, *MSR1*, *CD163* and phagosome proteolytic cathepsins, e.g., *CTSB*, *CTSL* and *CTSD* in the NACT samples. However, angiogenesis and ECM-regulation genes were downregulated in NACT samples (Fig. 1D, Supplementary Data 3). GSVA showed significant upregulation of phagocytic, complement pathways, antigen processing and presentation with upregulation of lysosomal and phagolysosome pathways in NACT samples (Fig. 1E).

Of particular interest was stabilin-1 (*STAB1*), a macrophage scavenger receptor that was expressed and upregulated in most of the clusters after NACT (Fig. 1F). STAB1 is a multifunctional scavenger receptor with intracellular sorting functions that performs phagocytic clearance of apoptotic cells via lysosomal degradation pathways[18–20]. STAB1 is marker for alternatively activated macrophages[21] and increased expression is associated with worse prognosis in several solid cancers[22,23].

We validated *STAB1* upregulation after NACT in bulk RNAseq from a separate cohort of sixteen HGSOC patients (four PDS, twelve NACT) that we previously published[24] (adjusted $p = 0.02$, log2FC = 1, Supplementary Data 3). Furthermore, we validated this finding at the protein level using archived sections of omental metastases from eighty HGSOC patients (twenty-two PDS and fifty-eight NACT). We found significant increase in CD68 and STAB1 positive cells in the NACT samples ($p = 0.04$ and $p = 0.0051$) (Fig. 1G). STAB1 positive macrophages were widely distributed in both stromal and malignant areas (Supplementary Fig. S1F). STAB1 levels were not significantly different between patients who received three versus six cycles of NACT or between BRCA mutated and HRD (BRCAm/HRD) tumours versus BRCA wild-type and non-HRD tumours (BRCA-WT) (Supplementary Fig. S1G). However, in all the subgroup analyses, STAB1 levels were significantly higher in the NACT samples compared to the PDS samples (Supplementary Fig. S1G, S1H). Looking at STAB1 expression change with treatment response, assessed by the Chemotherapy Response Score[25,26] (CRS), STAB1 positive cells were significantly lower only in CRS3 samples which had no residual tumour in the omentum (Supplementary Fig. S1G). Higher STAB1 levels in the NACT samples was also seen in adnexal tumours (Supplementary Fig. S1H). We confirmed the increase in STAB1 positive macrophages post-NACT in a cohort of forty-two patients with available patient-matched pre- and post-NACT samples ($p = 0.0003$) (Fig. 1I, 1J). The increase in STAB1 positive macrophages was seen in BRCA-WT and BRCAm/HRD tumours, however, was not significant in BRCAm/HRD group in the patient-matched cohort (Fig. 1J). Patients with high tumour macrophage STAB1 infiltrate post-NACT had significantly worse PFS and OS than patients with lower macrophage STAB1 levels ($P = 0.0046$ and $P = 0.043$) (Fig. 1K) but this was not significant for pre-NACT STAB1 levels. HGSOC patients from the ICGC dataset[27] with high enrichment scores of M0:STAB1 cluster genes had significantly worse survival than those with low scores (Supplementary Fig. S1I).

We concluded that there was an increase in macrophage infiltration and STAB1 expression in the post-NACT samples. We also found significant upregulation of pathways related to scavenger receptors, phagocytosis and antigen presentation but also upregulation of pathways suggestive of increased antigen degradation which could possibly reduce any anti-tumour immune effect of NACT.

We hypothesised blocking STAB1 could reduce lysosomal antigen degradation and improve antigen presentation after NACT which could repolarize macrophages towards an APC phenotype enhancing their anti-tumour role.

### NACT favours monocytic dendritic cell differentiation

We annotated 3,356 cells as dendritic cells (DC) in six distinct clusters (Supplementary Fig. S1J and S1K). Conventional DC clusters, cDC1 and cDC2, expressed *CLEC9A* and *CLEC10A* respectively, whereas mature DC (mDC) expressed *BIRC3*, *CCR7* and *LAMP3* genes. Monocytic DC expressed *CD207*, *CD1A* and *HLA-DRQ* and Plasmacytoid DC (pDC) expressed *LILRA4, IL3RA* and *JCHAIN* genes[28] (Supplementary Data 2). Individual patients had variable percentage of the dendritic cell clusters with no difference between PDS and NACT (Supplementary Fig. S1L, S1M).

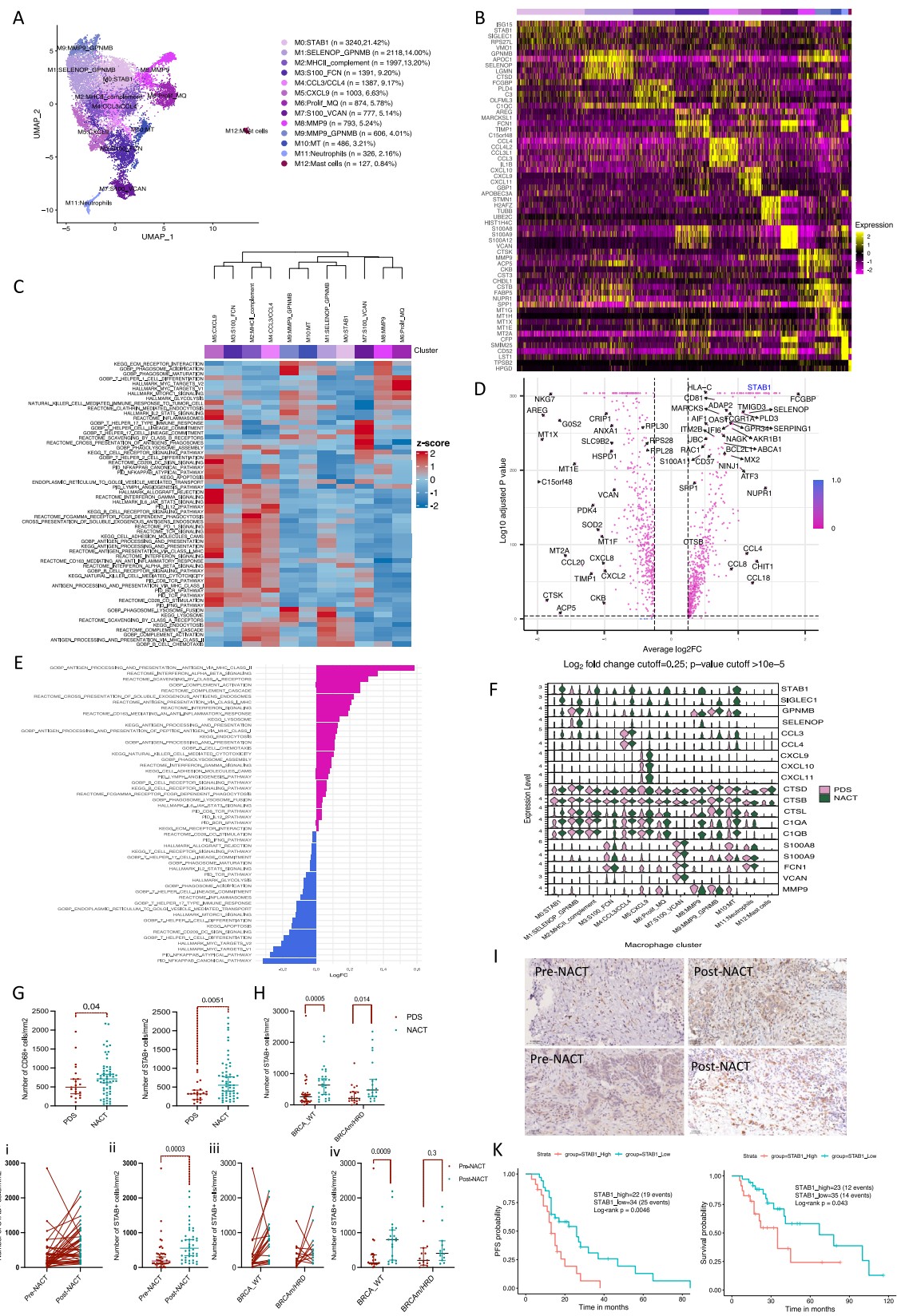

DC from NACT samples showed upregulation of antigen presenting pathways, but also upregulation of antigen degradation pathways such as lysosome and phagolysosome formation (Supplementary Fig. S1N and Supplementary Data 3).

## NACT increases Tregs infiltrate in tumour

We identified thirteen clusters of CD4 T cells (10,427 cells) (Fig. 2A). There were two CD4 naïve-memory clusters expressing *TCF7*, *CCR7* and *SELL* and two effector memory T cell clusters expressing *CD40LG*,

**Fig. 1 | Human myeloid cell subpopulations in HGSOC omental metastases.**
**A** UMAP showing integrated myeloid cell subpopulations ($n = 13$) from seven patients (3 PDS and 4 NACT). **B** Heatmap showing top five upregulated genes per cluster calculated using the default Wilcoxon sum rank test in Seurat package and arranged by log2FC. **C** Differentially enriched pathways in the macrophage clusters, GSVA analysis, coloured by z transformed mean GSVA scores. **D** Differentially expressed genes NACT versus PDS. Wilcoxon rank sum test. **E** Differentially enriched pathways NACT versus PDS using GSVA and two-sided unpaired limma-moderated $t$ test. **F** Violin plot of expression level of selected genes in PDS (pink) and NACT (green) in each macrophage cluster. **G** CD68 and STAB1 positive cells per mm² in HGSOC omenta (PDS = 22, NACT = 58 patients). Error bars median and 95% CI. Two-sided Mann–Whitney test. **H** STAB1 positive cells/mm² comparing PDS and NACT in both BRCA-wild type ($n = 42$) and BRCAmutated/HRD patients ($n = 27$). Two-sided Mann–Whitney test. **I** Immunohistochemistry staining for STAB1 positive cells in patient-matched omental samples pre- and post-NACT (n = 2 patients). **J** i-ii) STAB1 positive cells in patient-matched samples. 42 patients were included who had pre-treatment biopsy and biopsy after NACT. 36 of the pre-NACT samples were omental metastasis and 4 peritoneal biopsies, 1 cervical and 1 liver biopsies. iii and iv) Analysis of STAB1 positive cells/mm² in the patient-matched cohort comparing pre- and post-NACT in both BRCA-wild type ($n = 21$ paired samples) and BRCAmutated/HRD patients ($n = 13$ paired samples). Error bars at median and 95% CI. Two-sided Wilcoxon matched pairs signed rank test. **K** Kaplan–Meier survival and PFS curves (Mantel–Cox), STAB1-high versus low levels measured by IHC ($n = 58$ and 56 patients). 60% was the cut-off between STAB1 high and low levels.

---

*ANXA1*, *CD69* and *ZFP36*. There was one cluster with high expression of interferon stimulated genes (ISGs), one cluster of pre-dysfunctional and dysfunctional CD4 cells with high levels of *CXCL13* and *NMB* respectively. We identified five Treg clusters defined by *FOXP3* expression (Fig. 2B, Supplementary Data 2). All CD4 clusters were present in all patients at variable percentages (Supplementary Fig. S2A-S2B). Pathway enrichment analysis clustered CD4 T cells into the naïve, effector, dysfunctional and regulatory groups (Supplementary Fig. S2C).

Around 30% of CD4 T cells in the NACT group were naïve T cells. NACT significantly upregulated *FOXP3* and ISG genes (Fig. 2C, Supplementary Data 3). The activation and dysfunctional genes, such as *ICOS*, *PDCD1*, *TIGIT* and *LAG3*, were higher in Treg clusters than other CD4 clusters (Fig. 2D). In view of this, we studied the Treg clusters in more detail and found that Tregs constituted 26.7% of all CD4 T cells. CD4_Tregs1 and 2 clusters had an immune-suppressive signature with CD4_Tregs2 cluster expressing higher levels of *ICOS* and *PDCD1*. The other two clusters expressed ISGs and proliferation-related genes, respectively. Similar to Zheng et al.[29], we found one Treg cluster expressing effector CD8 cell genes.

Trajectory inference using Slingshot[17] predicted a trajectory connecting Treg clusters with pre-dysfunctional, effector and naïve memory T cell clusters (Fig. 2E). We also found that Treg clusters shared some TCR clonotypes with naïve memory, effector, dysfunctional T cell and other Treg clusters (Fig. 2F, S2D). We therefore hypothesised that the Tregs could be developmentally related to other CD4 T cells in the TME, which might have been repolarised into immunosuppressive Tregs, rather than natural/thymic Tregs.

There was an increase in CD3 and CD4 cell infiltrates in omental tumours post-NACT (eight PDS and twenty-three NACT patients) but this was statistically significant only for CD3 cells (Fig. 2G). We then studied two subtypes of CD4 T cells; Tregs and T helper cells using IHC on omental samples from eighty HGSOC patients (PDS = 22, NACT = 58 patients). We found significant increase in the number of FOXP3 positive cells in the NACT samples, compared to PDS samples, but no change in TBET positive cells (Fig. 2H). The number of FOXP3 positive cells was not different between patients' samples who received three versus six cycles of NACT or between BRCAm/HRD and BRCA-WT tumours (Supplementary Fig. S2E). However, in the subgroup analysis, FOXP3 positive cells were significantly higher in the NACT samples compared to PDS samples (Supplementary Fig. S2E, 2I). FOXP3 levels were significantly lower only in CRS3 samples (Supplementary Fig. S2E). FOXP3 positive cells were also significantly increased after NACT in adnexal tumours (Supplementary Fig. S2F). The increase in FOXP3 positive cells after NACT was further confirmed in a cohort of forty-two patients with matched pre- and post-NACT samples (Fig. 2J, 2K). The increase in FOXP3 positive cells in the NACT samples was seen in both BRCA-WT and BRCAm/HRD tumours, however, was not significant in the BRCAm/HRD group in the patients-matched cohort. Patients with high FOXP3 infiltrate post-NACT had significantly worse PFS than patients with lower FOXP3 infiltrate (Fig. 2L). This was not statistically significant for pre-NACT FOXP3 levels.

We did not find significant clonal expansion in CD4 cells with NACT (Fig. 2M) which could be due to non-specific T cell activation post-NACT, small number of samples studied or lack of patient-matched samples.

In conclusion, CD3 T cell infiltrate and FOXP3 positive cells were higher in the NACT samples. We hypothesised that targeting FOXP3, the master regulator of Tregs, might repolarise T cells into effector T cells or/and reverse their immune suppressive effect.

## NACT activates B cells
We identified seven clusters of B and plasma cells (2840 cells) (Supplementary Fig. S2G, Supplementary Data 2). Mature B cells (B.Mature) expressed *FCER2*; the B.ISG cluster expressed high ISG signature, two plasma cell clusters were characterised by high expression of *JCHAIN* and *MZB1* and proliferation genes. Cluster (B.GC) and (B.Prolif) most likely represented cells from the germinal centre of tertiary lymphoid structures due to the expression of *TCL1A*, *LMO2*, *LTB* and proliferation genes. B.GC and B.prolif clusters were only found in the NACT samples (Supplementary Fig. S2H, S2I).

Compared to PDS, there was significant upregulation of genes related to B cell maturation, antigen presentation and germinal centre formation in NACT samples (Supplementary Fig. S2J, Supplementary Data 3). There was clonal expansion and transition in B.prolif and plasma cell clusters but there was no clonal expansion with NACT (Supplementary Fig. S2K–S2M). We found a possible increase in B cells post-NACT ($p = 0.06$) using flow cytometry on omental samples from thirty-one HGSOC patients (Supplementary Fig. S2N).

## NACT effects on CD8 and NKTgd cells
We identified fifteen clusters of CD8 cells (24,155 cells) (Fig. 3A). Three clusters were naïve memory T cells, expressing *CCR7*, *SELL*, *TCF7* and *GZMK*, four cytotoxic CD8 T cell clusters expressing *NKG7* and *GZMH* with one of these cytotoxic clusters also expressing high level of the DC chemoattractant *XCL1*. The *CCL3L/CCL4L*-expressing cluster could interact with the inflammatory myeloid cells expressing the relevant receptors. The NKT cell cluster expressed the highest level of *GZMB*, *PERF1* and *GNLY* while predysfunctional and dysfunctional CD8 clusters expressed high levels of *CXCL13* and *PDCD1*, respectively. There was one cluster of proliferating CD8 cells and one of ISG-expressing CD8 cells (Fig. 3B, Supplementary Data 2). Each patient had variable percentage of each cluster except for one cluster (CD8_CXCR6) which comprised cells mostly from patient S2(PDS) (Supplementary Fig. S3A, B).

NACT samples showed significant upregulation of dendritic cell chemoattractant cytokines and cytotoxic genes e.g., *CCL3*, *CCL4*, *GZMB* (Fig. 3C) suggesting an increase in T cell cytotoxicity and myeloid cell engagement. Figure 3D shows the upregulated pathways with NACT.

Flow cytometry revealed a significant increase in CD8 T cells in NACT-treated HGSOC omental metastases of our thirty-one patients (PDS = 8, NACT = 23)(Fig. 3E).

We looked at the CD8 TCR in the different clusters and noted clonal expansion in dysfunctional CD8 T cell clusters and one cytotoxic

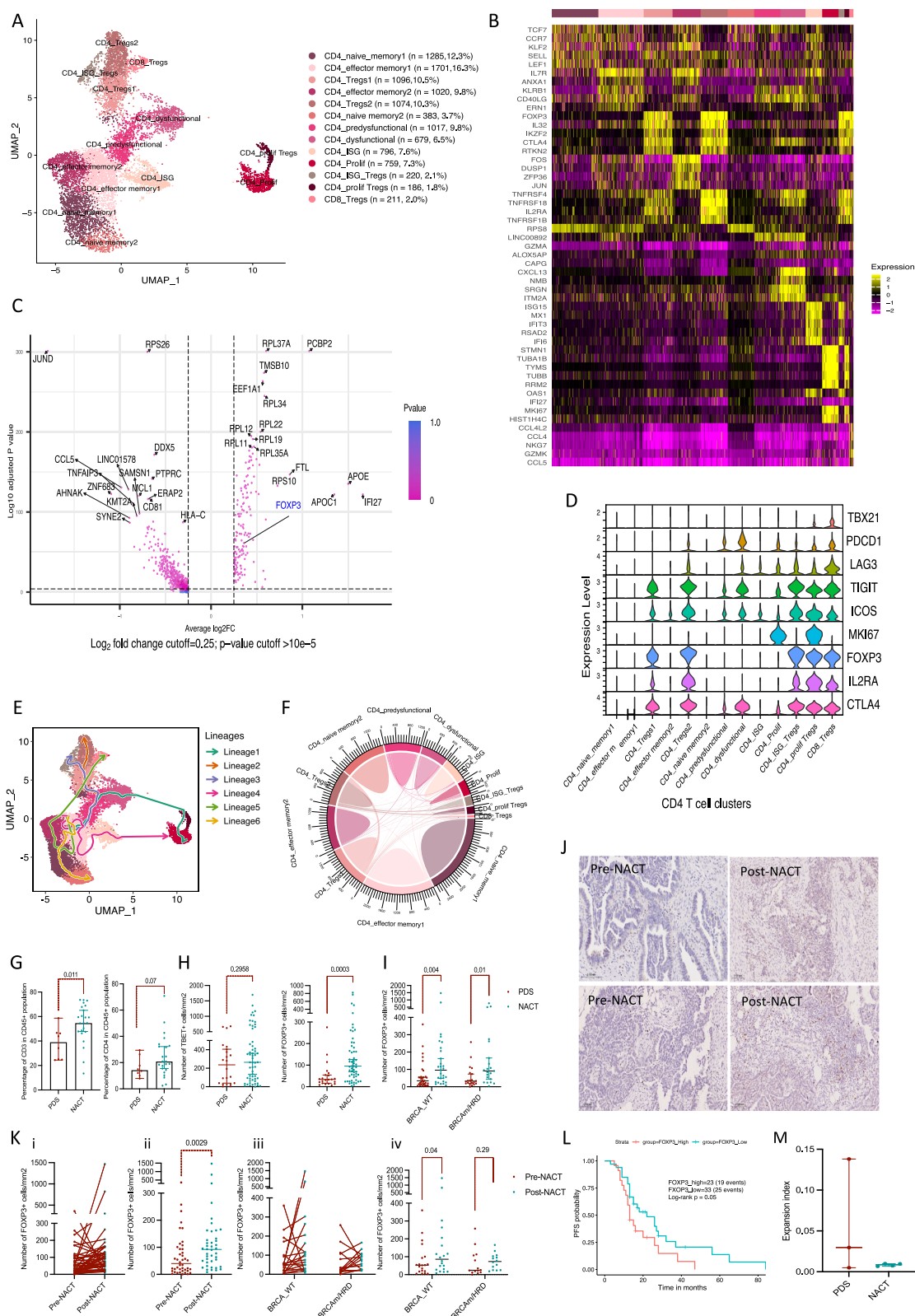

T cell cluster. CD8 clusters expressing myeloid chemoattractant molecules, dysfunctional and cytotoxic clusters had higher clonal transition index than in the other clusters (Supplementary Fig. S3C, S3D). However, similar to CD4 cells, there was no significant clonal expansion after NACT (Supplementary Fig. S3E).

We identified five subpopulations of NK cells (2780 cells) and four of T gamma delta cells (Tgd) based on the expression of CD3 genes and both gamma and delta chains (Fig. 3F). NK subpopulations were either cytotoxic (NK.SPON2), chemoattractant to dendritic cells (NK.KRT86) or exhausted NK.KRT86 (Fig. 3G, Supplementary Data 2). Individual samples had variable proportions of each cluster (Supplementary Fig. S3F). NACT increased the chemoattractant NK cells but not the cytotoxic ones (Supplementary Fig. S3G) and increased NK cell infiltration (Supplementary Fig. S3H).

**Fig. 2 | Human CD4 T cell subpopulations in HGSOC omental metastases.**
**A** UMAP showing integrated CD4 T cell clusters from seven patients, 4492 and 5935 cells pre- and post-NACT, respectively. **B** Heatmap showing top five upregulated genes per cluster. **C** Differentially expressed genes in NACT versus PDS. Wilcoxon rank sum test. **D** Violin plot showing the expression of different Treg and exhaustion genes in CD4 clusters. **E** Slingshot Trajectory analysis for CD4 cell subpopulations. **F** Circos plot showing TCR repertoire and clonal sharing in CD4 cell clusters coloured by the colour of the cluster. Width of chords reflects clonal sharing. **G** Percentage CD3 and CD4 T cells in CD45+ cells using flow cytometry on omental samples from HGSOC omena ($n = 8$ PDS, $n = 23$ NACT). Gating on singlet/live/CD45 + /CD3+ cells. Error bars at median and 95%CI. Two-sided Welch $t$-test. **H** FOXP3 and TBET positive cells/mm$^2$ in 80 HGSOC omena (PDS = 22, NACT = 58 patients). Error bars at median and 95%CI. Two-sided Mann–Whitney test. **I** FOXP3 positive cells/mm$^2$ comparing PDS and NACT in both BRCA-wild type ($n = 42$) and BRCAmutated/HRD patients ($n = 27$). Two-sided Mann–Whitney test. **J** IHC of human omentum staining for FOXP3 ($n = 2$ patients) pre- and post-NACT. **K** i and ii) FOXP3 positive cells/mm$^2$ in 42 patient-matched HGSOC samples. iii and iv) Analysis of FOXP3 positive cells/mm$^2$ in the patient-matched cohort comparing pre- and post-NACT in both BRCA-wild type ($n = 20$ paired samples) and BRCAmutated/HRD patients ($n = 14$ paired samples). Error bars at median and 95%CI. Two-sided Wilcoxon matched-pairs signed rank test. **L** Kaplan–Meier survival curves (Mantel–Cox) between FOXP3-high and low levels measured by IHC on NACT samples ($n = 58$). 60% was cut-off between FOXP3 high and low levels. **M** Expansion index of CD4 TCR in PDS and NACT studied by STARTRAC calculated as 1-normalised Shannon entropy. Error bars at median and 95%CI. Two-sided unpaired student $t$-test.

## Anti-stabilin1 (anti-stab1) antibody treatment delayed antigen degradation and improved T cell killing in vitro

To study the effect of anti-stab1 antibody on macrophages, we conducted a series of in vitro experiments to study phagocytic, antigen processing and secretory functions. Monocytes from healthy human PBMCs were cultured in the presence of recombinant human MCSF[30]. On day 5, macrophages were treated with isotype or anti-stab1 antibody, then on day 6 with dexamethasone and IL4 to induce STAB1 expression[31]; IFNγ for M1-like polarization; carboplatin or paclitaxel. Monocyte-derived dendritic cells (moDC) obtained by culturing monocytes in GMCSF, IL4 and IFNγ[32,33] acted as a negative control as they do not express STAB1. We cultured day 7 macrophages from the above conditions with Phrodo® coated dead cancer cells and imaged with the Incucyte S3. Macrophages phagocytosed the dead cells within few minutes, with a progressive increase in the red intensity due to the acidic PH of the phagolysosome, followed by a red signal decline due to degradation of the phagocytosed cells (Supplementary Fig. S4A). Blocking the STAB1 receptor did not alter phagocytic capacity of the macrophages (Supplementary Fig. S4B) but the red object mean intensity remained higher for longer in anti-stab1 antibody-treated macrophages compared to isotype control suggesting delayed antigen degradation (Fig. 4A-B, Supplementary Fig. S4A). moDC cells showed no change in phagocytic capacity or red signal intensity with anti-stab1 antibody treatment.

To study the effect of anti-stab1 antibody treatment on antigen degradation, we added DQ™-Ovalbumin to the macrophages and imaged using the Incucyte S3. DQ™-Ovalbumin emits green fluorescence on protease degradation in a pH-independent manner. Macrophages treated with anti-stab1 antibody had significantly lower antigen degradation than isotype-treated macrophages in all culture conditions (Supplementary Fig. S4C) to levels as low as dendritic cells which are known to have low level of antigen degradation[34]. Using LysoSensor™ green fluorescence dye, anti-stab1 antibody treated macrophages had significantly lower lysosomal acidity than isotype-treated macrophages at all culture conditions (Supplementary Fig. S4D). Flow cytometry showed that anti-stab1 treated macrophages had significantly lower lysosomal LAMP1 expression than isotype-treated macrophages (Supplementary Fig. S4E–S4F). Anti-stab1 treated macrophages had significantly higher TNFα and CD11C expression and lower PDL1 expression compared to isotype-treated ones suggesting macrophage repolarization after anti-stab1 antibody treatment (Fig. 4C).

Next, we tested the effect of conditioned medium from anti-stab1 antibody treated macrophages on T cell phenotype, activation and killing. T cells were cultured in macrophage supernatant from anti-stab1 or isotype antibody treated macrophages, activation nanobeads and IL-2 for 3 days. On day 4, activated T cells were harvested and co-cultured with live red lentivirus-transduced G164 cells and the plate was imaged in Incucyte S3. Alternatively, T cells were harvested for flow cytometry analysis. T cells cultured in conditioned medium from anti-stab1 antibody treated macrophages showed significantly better G164 killing than T cells cultured in conditioned medium from isotype-treated macrophages suggesting that the former had more T cell activating mediators (Fig. 4D-E). This was confirmed by flow cytometry showing higher Ki67 expression in both CD4 and CD8 cells, higher CD4 TBET expression and higher CD8 granzyme B expression in T cells cultured in supernatants from anti-stab1 antibody treated macrophages than isotype-treated macrophages (Fig. 4F).

We concluded that anti-stab1 antibody delayed antigen degradation, repolarised macrophages to CD11C expressing macrophages, increased TNFα secretion and changed macrophage cytokine profile to be more T cell activating. Importantly, the anti-stab1 antibody did not reduce apoptotic cell uptake, which has been linked to autoimmunity[35], but changed the way macrophages process the phagocytosed antigen, potentially strengthening the antitumour immune response.

## FOXP3-ASO treatment significantly reduced the percentage of FOXP3 expressing T cells and improve T cell killing in vitro

To target Tregs, we used a FOXP3 anti-sense oligonucleotide, FOXP3-ASO (AZD8701) which selectively knocks down FOXP3 in human and mouse Tregs and inhibits Treg functionality[36]. Pan T cells were isolated from healthy human PBMCs, activated with TransACT beads and IL-2. T cells were then treated with FOXP3-ASO or scrambled siRNA (CTL-ASO) in presence or absence of rhTGFβ to induce stable FOXP3 expression.

We studied T cell activation and proliferation from days 3–9. There was significant reduction in CD4 FOXP3 expression and higher CD4 T helper cells (TBET + ) with progressive increase in CD4 and CD8 proliferation over time that were significantly higher in FOXP3-ASO treated T cells compared to CTL-ASO treated cells. There was higher expression of perforin on days 3 and 5 with significant reduction in exhaustion markers (CTLA4 and LAG3) in both CD4 and CD8 cells (Fig. 4G). Most of those changes were maintained even when T cells were cultured with rhTGFβ (Supplementary Fig. S4G).

We also studied the change in the cytokine profile of T cells treated with FOXP3-ASO and found increase in IL2, IL5, CCL3/4, TNFα, IFNγ and CD40L release from in vitro activated T cells treated with FOXP3-ASO compared to CTL-ASO, even in the presence of rhTGFβ (Supplementary Fig. S4H)

Activated T cells treated with FOXP3-ASO showed significantly better killing of the G164 HGSOC cell line compared to CTL-ASO treated T cells (Fig. 4H,I and Supplementary Fig. S4I).

We concluded that the improved T cell killing with FOXP3-ASO treatment was most likely due to reduction in the percentage of Tregs and change in cytokine release leading to reduction of their inhibitory effects on CD8 T cells.

Collectively, these in vitro data showed that FOXP3-ASO treatment in vitro reduced the polarisation of naïve T cells into regulatory T cells, reduced T cell exhaustion and improved T cell activation, proliferation and killing.

To study the efficacy of these inhibitors in vivo we needed a syngeneic mouse model that would replicate the human immune cells at scRNA-seq level and chemotherapy response.

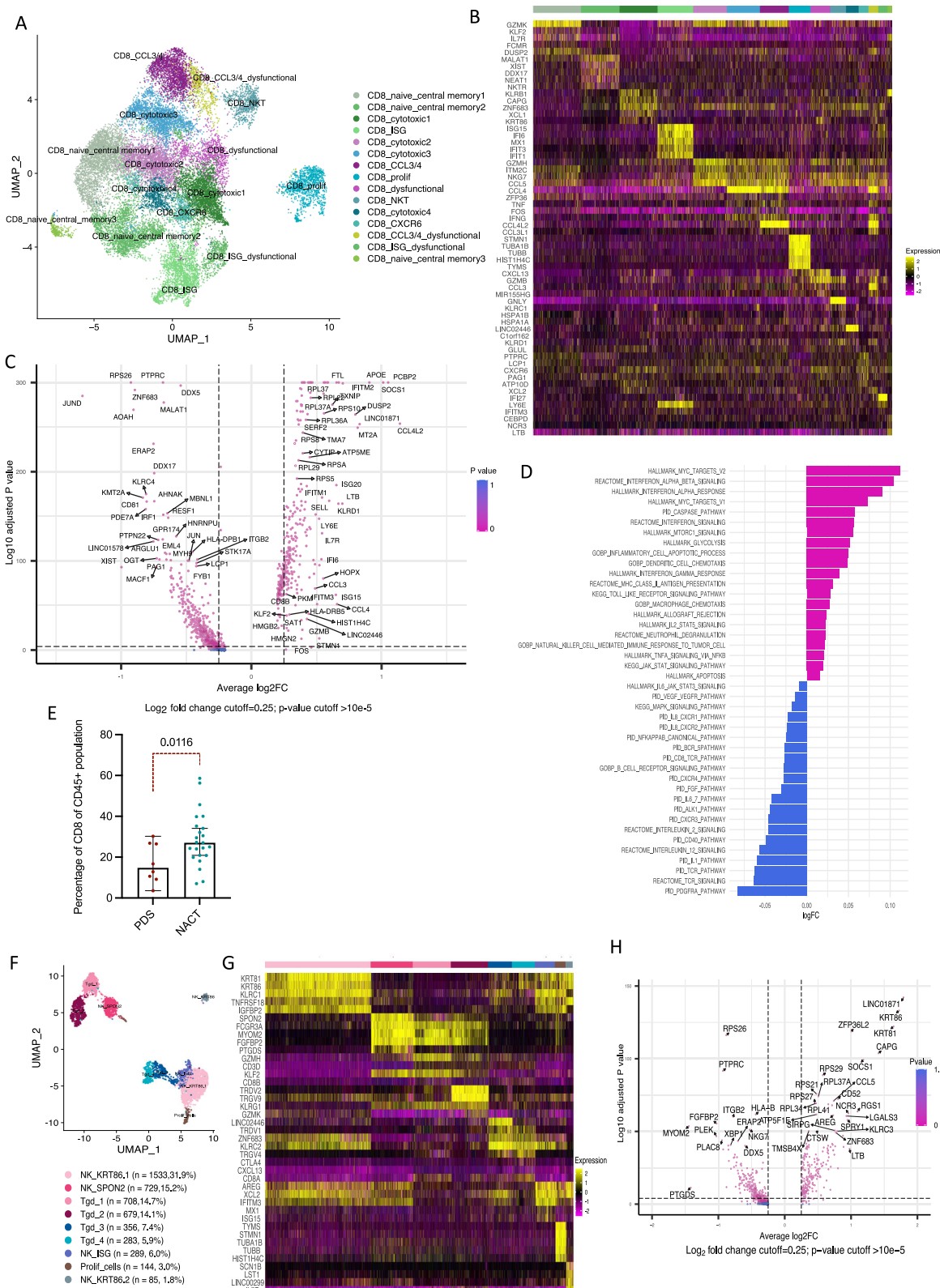

**Fig. 3 | Human CD8 T cell subpopulations in HGSOC omental metastasis.**
**A** UMAP showing integrated CD8 cell clusters from the above seven patients.
**B** Heatmap showing top upregulated genes per cluster. **C** Differentially expressed genes in CD8 T cells NACT versus PDS. Wilcoxon rank sum test. **D** Differentially enriched pathways in NACT versus PDS in CD8 cells. **E** Percentage of CD8 cells of total CD45 cells in 31 human omental samples ($n$ = 8 PDS, $n$ = 23 NACT) using flow

cytometry. Gating on singlet/live/CD45 + /CD3 + /CD8 cells. Error bars at median and 95%CI. Two-sided Welch $t$-test. **F** UMAP showing the integrated NKTgd cell clusters from 7 patients, 2780 and 2026 cells were annotated as NK and Tgd cells, respectively. **G** Heatmap showing top five upregulated genes per cluster in NKTgd subpopulations. **H** Differentially expressed genes in NKTgd cells in NACT versus PDS. Wilcoxon rank sum test.

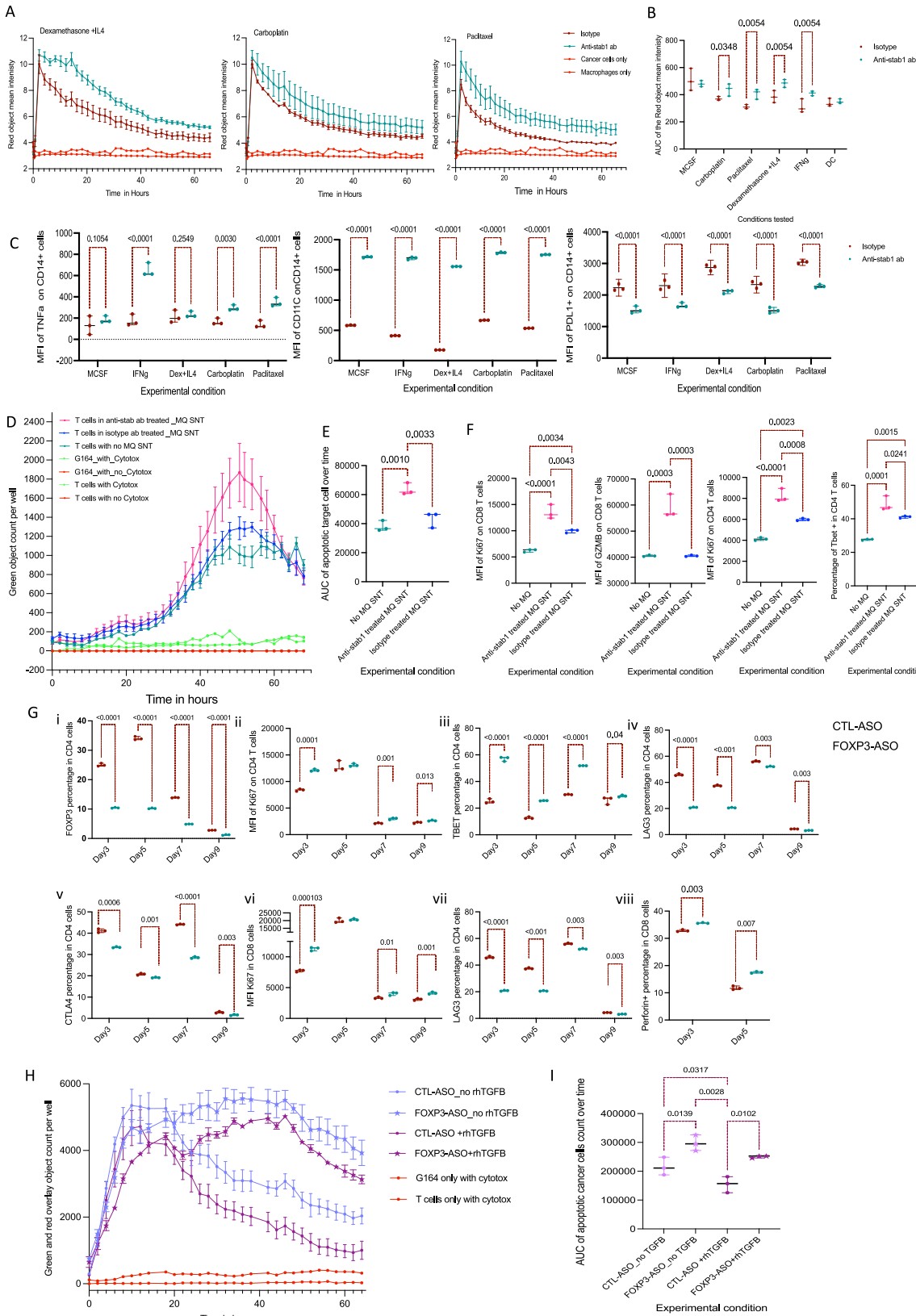

## HGS2 mouse model replicated human HGSOC disease in poor responders at single-cell level

We previously described a series of orthotopic syngeneic HGSOC mouse models which replicated many cellular and molecular aspects of human HGSOC[16]. Of these, we first chose to work with the HGS2 mouse model as this was poorly responsive to carboplatin[16] with a high EMT signature (Supplementary Fig. S5A). We treated mice with established HGS2 tumours with three cycles of carboplatin and paclitaxel (CT) to replicate NACT, harvesting omental tumours one week after the third dose. The entire tumour was processed for scRNAseq. After integration and filtering, we obtained 69,781 cells for further analysis: 39,000 and 30,781 from control and CT-treated mice

**Fig. 4 | The effect of anti-stab1 antibody and FOXP3-ASO treatment in vitro.**
**A** Incucyte human macrophage phagocytosis curves (*n* = 3 donors). Green = anti-stab1 antibody treatment, dark-red = isotype antibody treatment, red = macrophages alone and cancer cells alone (as negative controls). **B** Area under the curve (AUC) of the curves in 4 A (*n* = 3 donors). Error bars at median and 95%CI. Two-side paired *t*-test. **C** Median florescent intensity (MFI) of TNFα, CD11C and PDL1 on macrophages using flow cytometry. Gating on single/live/CD14+ cells. Error bars at median and 95%CI. Two-way ANOVA test with Tukey correction. **D** Incucyte S3 curves showing T cell killing (*n* = 3 donors) of G164 ovarian cancer cell line. T cells cultured in anti-stab1 antibody (pink) or isotype-treated macrophage conditioned medium (blue) or no macrophage medium (green). Green and red curves = T cells or cancer cells alone as controls for naturally occurring cell death in both. Error bars at median and 95%CI. **E** AUC of the curves in 4D (*n* = 3 donors). One-way ANOVA

with Tukey correction. **F** Flow cytometry analysis of T cells in 4D. Gating was done on live/singlet/CD4+ or CD8 + . One-way ANOVA with Tukey correction. **G** Time-course flow cytometry study of human T cells showing the effect of FOXP3-ASO over time (*n* = 3 donors). Gating was done on live/singlet/CD4+ or CD8 + . i) FOXP3 percentage of CD4, ii) MFI of Ki67 on CD4 + , iii) TBET percentage of CD4, iv) LAG3 on CD4, v) CTLA4 percentage on CD4, vi) MFI of Ki67 on CD8, vii) LAG3 percentage of CD8 cells and viii) Perforin+ cells in CD8. Error bars at median and 95%CI. Two-way ANONA t test with Tukey correction. **H** Incucyte S3 curves showing T cell killing of G164 cells plotted over time (*n* = 3 donors). Star=FOXP3-ASO treatment, circle= CTL-ASO treatment. Pink curves= T cells cultured without rhTGFβ. Purple curves= T cells cultured with rhTGFβ Error bars at median and 95%CI. **I** AUC of T cell killing curves in 4H (*n* = 3 donors). Error bars at median and 95%CI. Two-sided unpaired student t-test. rhTGFβ = recombinant human TGβ. CTL-ASO=control-ASO.

---

respectively. 19,655 cells were annotated as non-immune cells and 50,126 cells as immune cells. We focused on the immune cells to match the human data above.

## Murine myeloid cells replicated human macrophages in HGSOC omental metastases

We identified eleven clusters of monocytes/macrophages (12,436 cells) (Fig. 5A). Macrophage clusters replicated most of the human macrophage clusters, but with different proportions, with the predominant mouse macrophage phenotypes being ECM-regulating macrophages (M0:Mmp9, M3:Nrp2, M6:Tyrobp and M8:ECM/angiogenic) (Fig. 5B, Supplementary Data 4). We found monocyte/early macrophage clusters expressing *S100a8/9* genes and *Il1b and* other clusters expressing each of *Cxcl9*, *ISGs*, complement components, MHC class II and proliferation genes. Similarity analysis[37] with human macrophages (Fig. 5C) revealed a high cluster by cluster similarity. As with the human data, the clusters divided into two major groups, one related to antigen presentation, adaptive immune activation and chemoattraction and the other responsible for ECM and angiogenesis regulation (Fig. 5D).

Compared to non-treated mouse tumours, macrophages in CT-treated tumours showed significant upregulation of phagocytosis genes such as *Stab1, Mrc1, Mertk* and downregulation of ECM and angiogenesis such as *Mmp9* and *Col1a1* (Fig. 5E). Pathway enrichment analysis supported the above finding and showed upregulation of antigen presentation and adaptive immune activation pathways after CT. As with the patient tumours, there was significant upregulation of antigen degradation pathways (Supplementary Fig. 5B).

We validated the increase in omental tumour STAB1 expression in CT-treated mice using flow cytometry on CT-treated and control mice (Fig. 5F, Supplementary Fig. S5C), concluding that mouse macrophages replicated most of the features of human macrophages clusters and response to CT with upregulation of *Stab1*.

Four clusters of DC (488 cells) also showed high similarity with human DC (Supplementary Fig. S5D–S5F and Supplementary Data 4). CT treatment upregulated phagocytosis and antigen presenting pathways which was counterbalanced by upregulation of pathways suggesting enhanced antigen degradation (Supplementary Fig. S5G).

## Murine T and B cells showed marginal response to chemotherapy

We identified seven CD4 clusters (10,826 cells) (Fig. 5G, supplementary Data 4) which included naïve, effector and regulatory T cells. Supplementary Data 4 shows the differentially expressed genes in each cluster. Again, there was high similarity between mouse and human CD4 T cell clusters (Fig. 5H). Tregs represented 15.8% of the total CD4 T cells in four clusters similar to human clusters: two immune-suppressive clusters, ISG-Tregs and proliferating Tregs (Fig. 5I,J). Slingshot analysis suggested a trajectory starting at the naïve CD4 T cells and ending at Tregs (Fig. 5K).

We identified nine clusters of CD8 T cells (11,871 cells) and four clusters of NKTgd cells (783 cells) which showed high similarity to their human counterparts (Supplementary Fig. S6A–S6C and Supplementary Data 4). There was significant upregulation of T cell activation, T helper-2 and T helper-17 and Tregs differentiation and downregulation of T helper-1 pathways in CT-treated mice (Supplementary Fig. S6D). However, there was hardly any clonal sharing between the clusters (Supplementary Fig. S6E).

There were twelve clusters of B cells (9738 cells) (Supplementary Data 4). CT-treated B cells had significant upregulation of germinal centre formation and mature B cell pathways (Supplementary Fig. S6F). B cells showed clonal expansion and sharing to a greater extent than T cells (Supplementary Fig. S6G, H). As with human tumours, there was significant increase in both CD3 and CD19 infiltration after CT-treatment (Supplementary Fig. S6I).

In summary, the HGS2 mouse model largely replicated the immune cell clusters identified in human omental metastases and the effects of chemotherapy.

## Foxp3 inhibition and/or blocking STAB1 improved response to chemotherapy and prolonged survival in two syngeneic mouse models

We treated HGS2 mice bearing established peritoneal disease with anti-stab1 antibody or Foxp3-ASO (murine surrogate of AZD8701) as monotherapies and in combination with CT. The experiment, conducted twice, comprised control, CT, anti-stab1 antibody as monotherapy or with CT, Foxp3-ASO as monotherapy or with CT, and a triple combination arm with CT, anti-stab1 antibody and Foxp-3-ASO. The mice did not show any signs of autoimmunity or other adverse effects with any treatment. Figure 6A shows a Kaplan–Meier survival curve of the two experiments pooled together with individual experiments shown in Supplementary Fig. S7A–D.

The combination of anti-stab1 antibody with CT resulted in significant survival improvement over CT alone (98 days versus 83 days respectively, *P* = 0.0001), with three of fifteen mice as long-term survivors. Furthermore, Foxp3-ASO in combination with CT resulted in statistically significant survival improvement over CT alone (145 days versus 83 days respectively, *P* < 0.0001) with five of seventeen mice surviving beyond 250 days.

The triple combination of Foxp3-ASO with CT and anti-stab1 antibody improved survival over CT alone and CT plus anti-stab1 antibody (median survival 200 days compared to 83 days and 98 days respectively, *P* = 0.0001) and CT plus Foxp3-ASO (median survival of 200 days compared to 145 days, respectively, not statistically significant) with seven of sixteen mice surviving beyond 250 days in the triple combination treatment arm.

Anti-stab1 antibody monotherapy resulted in small but statistically significantly survival benefit (median survival of controls 62 days versus 72 days *P* = 0.0004). Foxp3-ASO monotherapy and combination with anti-stab1 antibody did not result in survival benefit.

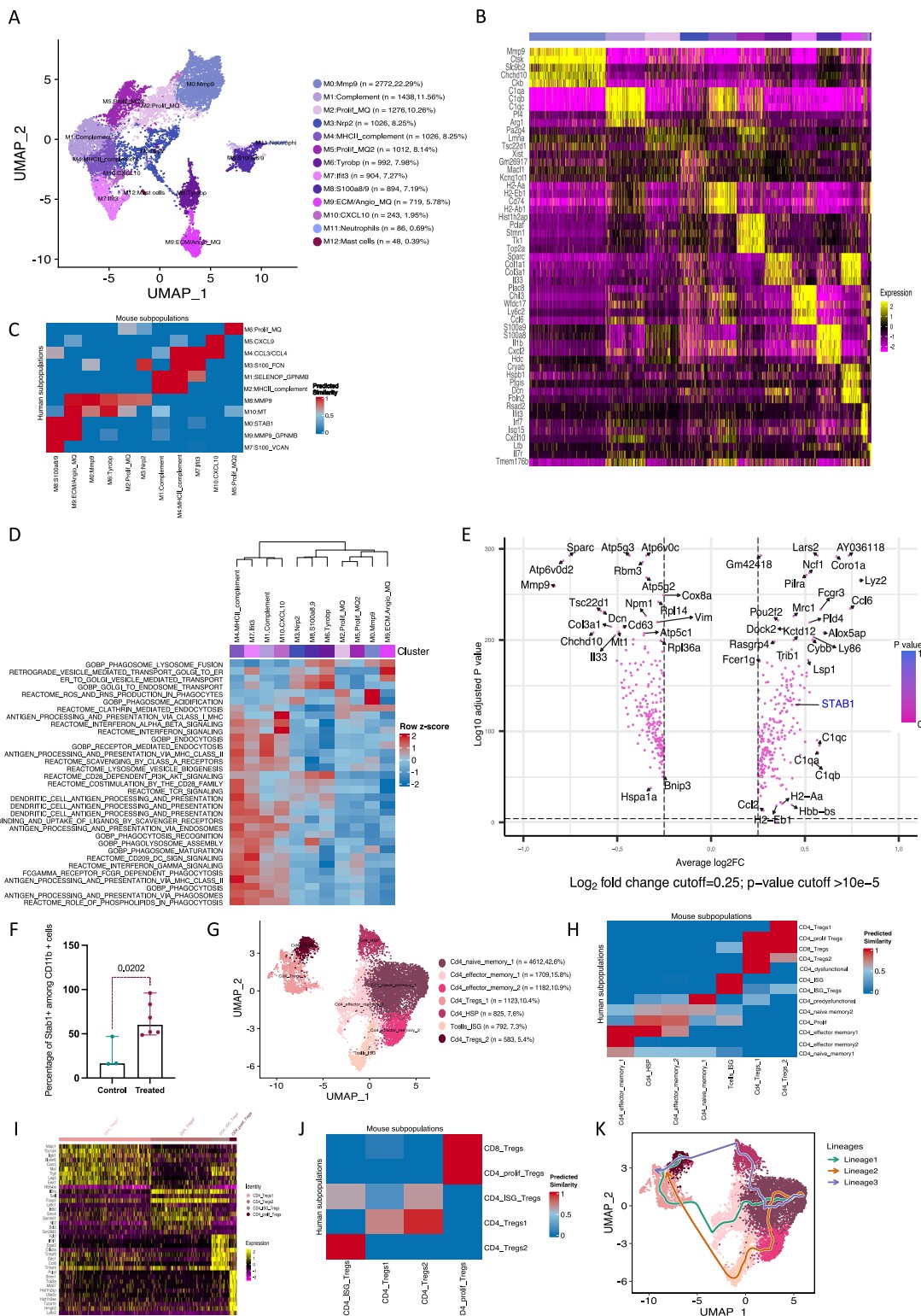

**Fig. 5 | Myeloid subpopulations in HGS2 omental tumours. A** UMAP of integrated mouse macrophage subpopulations (*n* = 13) (3 control and 3 CT-treated mice). **B** Heatmap of top five upregulated genes per cluster. **C** Similarity analysis between HGS2 and human myeloid cell clusters. **D** Differentially enriched pathways in the macrophage clusters using GSVA analysis, coloured by z transformed mean GSVA scores. **E** Differentially expressed genes in CT-treated compared to control mice. Wilcoxon rank sum test. **F** Percentage of STAB1 positive cells in HGS2 omental tumour macrophages using flow cytometry. Gating was done on singlet/live/

CD45 + /CD11b + /Ly6C_G neg/F4_80 + /STAB1+ positive cells on omental samples from 3 control and 6 CT-treated mice. Error bars at median and 95%CI. Two-sided unpaired *t*-test. **G** UMAP of integrated mouse CD4 subpopulations (*n* = 7), from control and CT-treated mice (*n* = 3 each). **H** Similarity analysis between HGS2 and human CD4 cell clusters. **I** Heatmap of top upregulated genes per Treg cluster. **J** Similarity analysis between human and mouse Treg cell clusters. **K** Slingshot trajectory analysis of CD4 T cells. CT=carboplatin and paclitaxel.

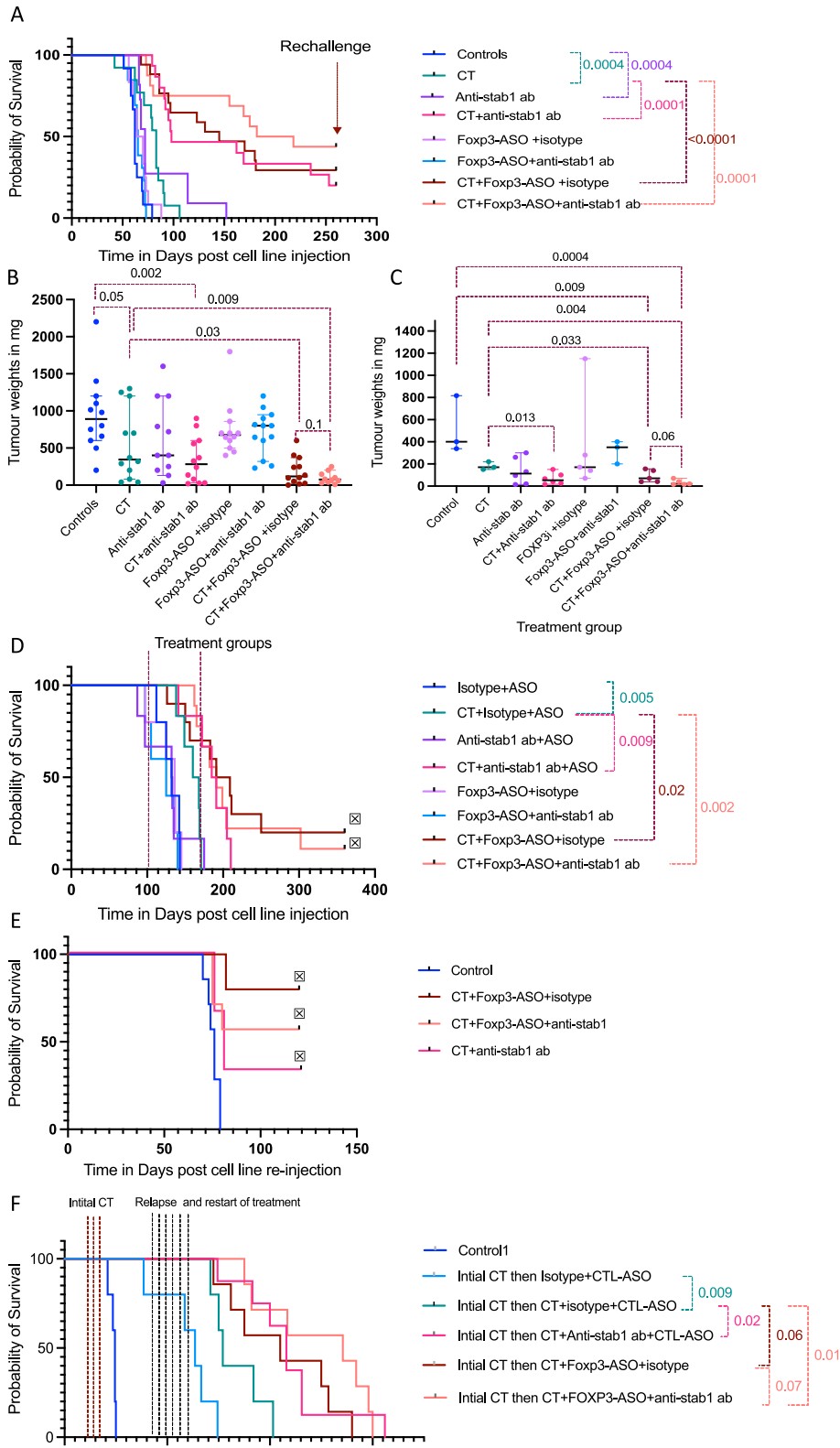

Peritoneal tumour weights at humane endpoint were significantly less in the CT plus anti-stab1 antibody treatment arm, CT plus Foxp3-ASO arm and with the triple combination compared to controls or CT alone (Fig. 6B).

In a further experiment with the same treatment arms, we harvested tumours at 70 days. Tumour weights in CT plus anti-stab1 and/or Foxp3-ASO treatment arms were significantly less than controls and CT treatment arms (Fig. 6C).

We tested the same treatment combinations on established peritoneal disease in a second non-EMT HGSOC syngeneic mouse model, 30200[16] (Supplementary Fig. S5A). There was significant survival advantage with CT over controls (median survival, 164 days versus 132 days, $P = 0.005$). Anti-stab1 antibody plus CT further improved survival compared to CT (188 days versus 164 days, $P = 0.009$). Combining CT with Foxp3-ASO or CT plus Foxp3-ASO plus anti-stab1 antibody significantly improved survival over CT alone

**Fig. 6 | The effect of anti-stab1 antibody and Foxp3-ASO treatments on HGSOC mouse models survival and relapse. A** Kaplan–Meier HGS2 model survival curve (Mantel–Cox) comparing Foxp3-ASO and anti-stab1 antibody as monotherapies or in combination with CT. Data pooled from two experiments. $n = 5$–10 mice per group in each individual experiment. The red arrow indicates 2nd cell line injection (rechallenge) at around 260 days after the 1st cell line injection. **B** Tumour weights for the mice culled at humane endpoint, (excluding the long-survivors that were re-challenged), from the pooled experiment mention in A ($n = 10$–13 mice per group). Error bars at median and 95%CI. Two-sided P value for unpaired student $t$ test. **C** Time point experiment ($n = 3$–6 mice per group). Treatment was started at week 6 and tumours were harvested at day 70. Error bars at median and 95%CI. Two-sided $p$ value for unpaired student $t$ test. **D** Kaplan–Meier survival curve (Mantel–Cox) of 30200 syngeneic mouse HGSOC model. Dotted lines represent the start and end of treatment, 5–10 mice per group. **E** Long-term survivors for CT plus anti-stab1 antibody, CT plus Foxp3-ASO or triple combination ($n = 3$, $n = 5$ and $n = 7$, respectively) were rechallenged with HGS2 cell line 250–260 days after first cell line injection (experiment presented in (**A**)). Nearly age-matched control mice were

included ($n = 6$, 3 mice each experiment). Mice were culled when they developed clinically detectable tumour or 120 days after the second cell line injection. Symbol denotes the end of the experiment by culling the mice that did not develop clinically detectable tumour. CT=carboplatin and paclitaxel. **F** Kaplan–Meier 60577 murine model survival curve (Mantel–Cox) comparing Foxp3-ASO and anti-stab1 antibody as monotherapies or in combination with CT compared to standard CT for treatment of relapsed disease. In this experiment, all mice apart from control group received three doses of carboplatin and paclitaxel (3 dotted red vertical lines). Then the treated mice were randomised into 5 arms, 7–8 mice each, and left to develop relapse which was clinically diagnosed by development of ascites. At 85 days after cell line injection (which is 6 weeks after end of third CT cycle), treatment was restarted for 6 weeks (denoted by black vertical line) as follows, one arm received placebo control-ASO and isotype anti-body (light blue curve) and the remaining four arms received chemotherapy alone (green curve), CT plus anti-stab1 antibody (pink curve), CT plus Foxp3-ASO (dark red curve), CT plus both Foxp3-ASO and anti-stab1 antibody (orange curve), respectively.

---

(200 days and 195 days versus 164 days, respectively, $P = 0.01$ and 0.001).

### Tumour rejection in the long-term survivors after rechallenging with the same cancer cell line

The long-term survivors from the HGS2 mouse experiments described above were reinjected with HGS2 cells 250–260 days from the first HGS2 injection, after the absence of detectable tumours was confirmed clinically and radiologically. Control mice of similar age which had not had any previous procedures were included (Fig. 6E). Control mice reached humane endpoint at 59–75 days, matching the endpoint of the young control mice above. The experiment was terminated 120 days from the second cell line injection (approximately 380 days from the first cell line injection). Four of five long-term survivor mice in the CT plus Foxp3-ASO, four of seven mice in the triple combination arm and one out of three mice in the CT plus anti-stab1 antibody treatment arm were alive at 380 days with no grossly visible tumour post-mortem. This suggested that combining CT with our selected Tregs and/or macrophage targeting developed immune memory that prevented or delayed relapse. We looked at memory T cells in the omenta of rechallenged surviving mice and found large number of CD103 positive memory T cells compared to the age-matched controls with tumours (Supplementary Fig. S7E).

### Treatment of relapsed HGSOC after initial response to standard chemotherapy

As patients with HGSOC frequently relapse after first-line treatment, we used a third HGSOC syngeneic mouse model, 60577 that has a good initial response to chemotherapy then spontaneously relapsing after approximately 10 weeks. Mice responded to the first CT treatment with median survival of 127 days treated versus 49 days in control mice but CT treatment of relapsed disease gave a survival advantage of only 27 days (154 versus 127 days, $p = 0.009$) (Fig. 6F). Treatment of relapsed HGSOC with CT plus anti-stab1 antibody gave a significant increase survival compared to CT alone (median survival of 216 days compared to 154 days, $p = 0.02$). Treatment with chemotherapy plus Foxp3-ASO +/− anti-stab1 antibody was significantly higher than CT (210 days and 270 days compared to 154 days, $p = 0.06$ and 0.01).

### Anti-stab1 antibody treatment reduced antigen degradation and improved antigen presentation in vivo in the HGS2 model

To understand the effect of each of the above drug combinations, tumours from the experiment in Fig. 6C were processed for bulk RNA sequencing. Compared to control, CT significantly upregulated *S100a8/9, Fcna, Ctsd, Ctsb, Ctsl, Cxcl2* and *Mertk* (Supplementary Data 5) suggesting increased monocyte recruitment and

phagocytosis. However, CT significantly upregulated *Trem2*, characteristic of immunosuppressive macrophages (Supplementary Data 5) which was also detected in both human and mouse scRNA-seq data.

Compared to controls, anti-stab1 antibody treatment as monotherapy resulted in significant upregulation of antigen presentation pathways, TCR signalling, and allograft rejection compared to control mice (Supplementary Fig. S8A).

The combination of anti-stab1 with CT led to downregulation of phagosome and lysosome assembly, acidification and maturation pathways which could lead to reduced antigen degradation and improved antigen presentation (Fig. 7A). This agrees with the data presented above in Fig. 4A, B and S4B–F in which blocking STAB1 receptor with anti-stab1 antibody did not reduce phagocytosis of dead cancer cells but delayed antigen degradation.

### Foxp3-ASO as monotherapy and in combinations significantly stimulated anti-tumour response

Compared to control, Foxp3-ASO antibody monotherapy resulted in significant upregulation in immune activation pathways (Supplementary Fig. S8B, Supplementary Data 5). Moreover, when combined with CT, resulted in significant immune activation compared to CT (Fig. 7B). The triple combination resulted in significantly higher levels of immune activation pathways compared to CT alone or in combination with either anti-stab1 antibody or Foxp3-ASO (Fig. 7B, C).

### Correlating bulk RNAseq data to the immune cell subpopulations identified by single-cell RNAseq

We used the top upregulated genes from different immune subpopulations in the single cell dataset to generate subpopulation signatures. Using GSVA, an enrichment score was obtained for each treatment arm and the z transformed mean scores of all replicates were plotted. CT plus anti-stab1 antibody treatment arm showed the highest enrichment score for CXCL9 macrophage cluster and MHCII_complement cluster. The triple combination therapy had significant enrichment of subpopulations CD8_effector cell, cytotoxic T cells and matureDC (Supplementary Fig. S8C–F).

We then studied the global change in the TME using Ecotyper[38]. The tool uses the concept of CIBERSORTX to deconvolute bulk RNA seq data using a subpopulation signature derived from published scRNA-seq of human tumours. Based on the subpopulations identified, a carcinoma ecotype (CE) score is given on a scale of CE1–10. Higher scores were associated with better prognosis and patient survival. The controls and Foxp3-ASO plus anti-stab1 antibody were CE1–2, progressing to CE1–6 in CT treated arms and CE5–10 in Foxp3-ASO monotherapy, CT+Foxp3-ASO, CT+anti-stab1 and the triple combination (Supplementary Data 5). Thus, combining immunotherapies with

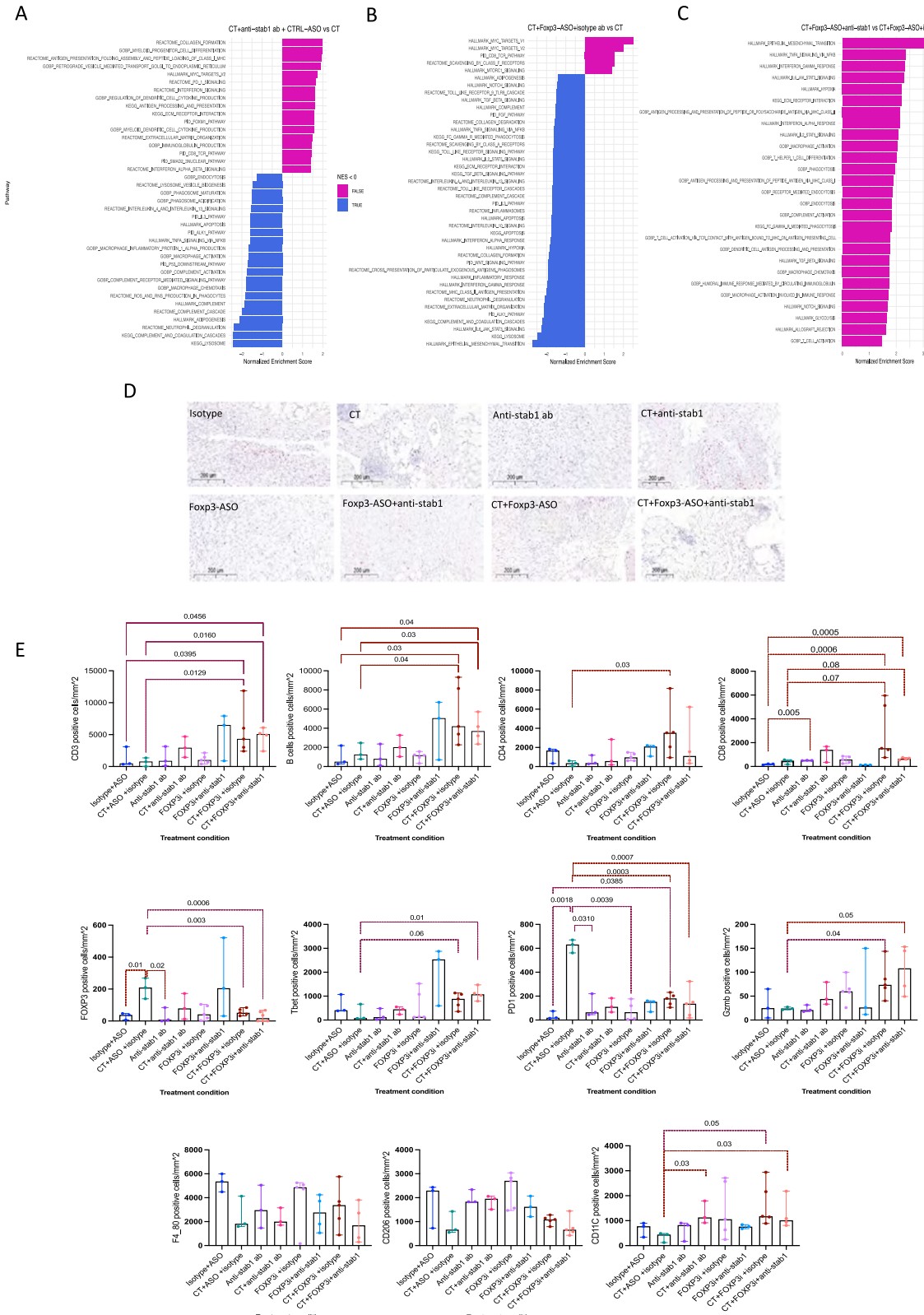

**Fig. 7 | The effect of different treatment combinations on murine HGS2 omental tumours. A**–**C** Bulk RNAseq was conducted on tumours from the time-point experiment in Fig. 6C. **A** Differentially enriched pathways in mice treated with CT plus anti-stab1 antibody compared to CT alone using Fgsea ($n = 3$ mice per group), $P < 0.05$ was considered significant. **B** Differentially enriched pathways in CT plus Foxp3-ASO compared to CT using Fgsea ($n = 5$ for CT plus Foxp3-ASO and 3 mice for CT), $p < 0.05$ was considered significant. **C** Differentially enriched pathways in the triple combination (CT plus Foxp3-ASO and anti-stab1 antibody) compared to CT plus Foxp3-ASO using Fgsea ($n = 5$ for the triple combination therapy and $n = 5$ for CT plus Foxp3-ASO) $p < 0.05$ was significant. **D** Representative IHC images for FOXP3 staining in HGS2 mouse omental tumours (scale bar at 200 µm). **E** Positive cells per mm² for immune infiltrate in different treatment arms in the time-point shown experiment in Fig. 6C ($n = 3$–5 mice in each group). Error bar at median and 95%CI. Two-sided unpaired student $t$-test (+/− welch correction).

CT resulted in a change in the TME that was associated with good prognosis in cancer patients.

Finally, using IHC we found significant decrease in the number of FOXP3 expressing cells in all the Foxp3-ASO treatment arms in both the tumour and the spleen (Fig. 7D, E, Supplementary Fig. S8G, S8H) and an increase in TBET expressing cells in the tumour (Fig. 7E). There was also a significant increase in the infiltration of T cells and B cells in CT plus Foxp3-ASO and the triple combination. There was no significant change in the number of macrophages but significant increase in the number of CD11C expressing antigen presenting cells with the combination therapy arms compared to chemotherapy (Fig. 7E).

## Discussion

Previously published scRNA-seq data from HGSOC tumours were mainly obtained from treatment naïve patients[10,39,40]. The only study on chemotherapy-treated patient samples focused on stromal effects[41].

Here, we showed the correlation between phagocytosis and chemotherapy in HGSOC patients and its implication on anti-tumour immune response. One key finding was upregulation of STAB1 in macrophages after carboplatin and paclitaxel, drugs that are widely used in treatment of cancers. STAB1 knock out in a murine melanoma cancer models reduced tumour growth and metastasis[42] and activated CD8 T cells similar to anti-PD1 treatment and the combined treatment had a more enhanced effect[43]. In agreement with those studies, we showed a survival benefit of anti-stab1 antibody in three murine HGSOC models when combined with chemotherapy and described its potential mechanism of action. We have not included immune checkpoint blockade, ICB, treatment in any experimental combinations due to the low expression of PD1 in advanced HGSOC[40] relative to other types of cancer.

We did not find significant effect of chemotherapy on the T cell and B cell clonal repertoires. However, this was hampered by the small number of samples that were only site-matched but not patient-matched. We used RNA as template as this limits the amplification to only the functional expressed receptor chains rather than whole repertoire genes. Jiménez-Sánchez et al[2] found significant increase in clonal expansion after NACT in site-matched samples but not in site-unmatched. Lee at al.[44] found significantly higher TCR diversity in PDS than NACT but no effect on clonal expansion. Both publications used DNA based methods for generating the TCR repertoire which correlate more with the T cell infiltrate rather than activity.

The concept of Tregs plasticity was described before in breast cancer[45] and glioblastoma[46]. In this study, we have shown that it may also exist in HGSOC and shown its potential clinical benefit. There are different ways of targeting Tregs e.g., CXCR4 antagonists[47] and anti-CD25[48]. We chose to use FOXP3-ASO(AZD8701) as it selectively knocks down FOXP3, the main Treg transcription factor, modulating Treg immune-suppressive activity without affecting the activation of other T cells. Revenko et al.[36] have recently shown that AZD8701 treatment increased CD8 T cell activation and anti-tumour effects in syngeneic mouse tumour models. This agreed with our in vitro results where we showed efficient reduction in FOXP3 expression even under TGFβ challenge, with improvement in T cell killing.

One of the limitations of this study is that we performed scRNAseq on a small number of unpaired human omental tumour samples which could hamper drawing firm conclusions on the immune effect of NACT especially with the known heterogeneity within and between patients. We also did not study the non-immune effects of chemotherapy or the spatial context. We used our scRNAseq dataset in an exploratory way and the results that we used to build our translational work were validated in several ways, e.g., by immunohistochemistry and flow cytometry on a larger cohort of eighty patients, forty-two of them had patient-matched samples. Using bulk RNAseq, Jimenez-Sanchez et al.[2] showed that chemotherapy-induced immune changes were more significant in site-matched samples than not in site-unmatched samples even if the site-unmatched samples were patient matched samples. Hence, to ameliorate the potential effect from using patient-unmatched samples, we focused on one metastatic site to be able to make meaningful comparisons and validated our findings in adnexal tumours.

The two immune cell molecules we identified have recently been tested in phase I clinical trials. FOXP3-ASO (AZD8701) was in Phase 1 Clinical trial (NCT04504669) in patients with advanced solid tumours. MATINS (NCT03733990 study) was the first-in-human study in metastatic solid tumours using a humanized anti-stab1 antibody, FP-1305 (bexmarilimab). Bexmarilimab was well tolerated up to a dose of 10 mg/kg[49] with no recordered dose-limiting toxicity. In spite of the low objective response-rate by RECIST criteria, there was promising disease control especially, in melanoma, but no responses in the ovarian cancer cohort[50]. The data we present here, suggest that the combination of Bexmarilimab with chemotherapy may lead to better response especially when the adaptive immune response is concomitantly targeted. Hence, we believe our work has a translational potential as it represents the preclinical work for possible successful treatment combinations, especially the triple combination. Both our immune targets are expressed in BRCA mutated/HRD and non-HRD tumours and our findings could be relevant to other human cancers expressing high levels of STAB1[23] and Tregs or showing upregulation after platinum-based chemotherapy.

## Methods
### Clinical samples

Human samples used in this research were obtained from Royal London Hospital under SIGNPOST (Systematic GeNetic Testing for Personalised Ovarian Cancer Therapy) study ethics (REC reference: 17/LO/0405) and from St George's University Hospital NHS trust under Barts Gynaecology tissue bank ethics (REC reference: 15/EE/0151) and an existing material transfer agreement (MTA). Details of the included patients are listed in Supplementary Data 1. Twenty-two patients had PDS and fifty-eight had NACT followed by debulking surgery. All patients were diagnosed with stage IIIC-IV high grade serous ovarian cancer. PDS samples were collected during primary debulking surgery (n = 22). NACT samples were collected during interval debulking surgery (n = 58), 3 weeks after the last chemotherapy cycle from patients. Patients received either carboplatin and paclitaxel or carboplatin and paclitaxel plus bevacizumab as NACT, (n = 50 and n = 8, respectively). Forty-six patients received three-four cycles of NACT, and twelve patients received five-six cycles of NACT.

Pre-NACT samples were collected from diagnostic biopsies before the start of treatment (n = 42). Clinical data was collected regarding the age at diagnosis, final histology, response on CT scan after NACT as assessed by RECIST criteria (stable, partial, progressive disease), Chemotherapy Response Score[25,26] (CRS1 = majority of the tumour in the omentum is still viable, CRS2 = evidence of both tumour apoptosis, fibrosis and viable tumour, CRS3 = no viable residual tumour in the omentum), Date of death or last clinic visit (data was available for 78 patients), date of relapse disease as evidenced by CT scan (data was available for 73 patients in total), chemotherapy regimen in NACT, optimal debulking achieved or not (residual disease) and BRCAm/HRD status.

### Human cell lines

AOCS1 HGSOC cell line was a kind gift from Professor David Bowtell and was grown in RPMI + Glutamax (cat.no 31331-093, Gibco) + 10% FBS + 100µg/ml pen/strep (cat. no. 15140-122, Gibco). Cells were incubated at 37 °C, 95% humidity and 5% CO2. Medium was replaced every 48–72 h. Cells were detached using 0.5% trypsin- EDTA (cat. no. 15400-054, Gibco) diluted 1:10 in PBS.Cells were authenticated by

short tandem repeat sequencing with the ATCC at the beginning of the project. Cell lines undergo regular monthly testing for mycoplasma.

IncuCyte NucLight Red Lentivirus reagent (Essence bioscience, Cat no.: 4475) was used to induce a stable expression of the red mKate2 protein in the cancer cells. The recommended multiplicity of infection (MOI) ranges between 3 to 6. The viral titre was $11.47 \times 10^6$ transduction units (TU) /ml (Lot number: LSP032219.02-051019). The lentivirus was used at MOI of 5 and polypyrene 8 µg/ml was used to increase the transduction efficiency. This was followed by puromycin selection at dose of 2 µg/ml.

## Mouse models

We used three syngeneic mouse models, HGS2, 30200 and 60577 which we published before[16]. HGS2 cancer cells were developed from C57/Bl6 mice but 30200 and 60577 on an FVB background. C57BL/6 wild type mice (aged 7 weeks) were purchased from Charles River Laboratories and FVB mice were purchased from Janvier Lab. Mice were acclimatised for 1–2 weeks prior to the initiation of the studies. Mice were housed under sterile conditions then in individually ventilated cages, a maximum of 5–6 animals per cage, fed with standard chow diet and water ad libitum, and maintained on an automatic 12 h light cycle at 22–24 °C. All studies were conducted using sterile techniques in accordance with the guidelines of the Animal Care Committee, Project License 70/7411, superseded by PP5394401, and personal licence PBE3719B3. Supplementary wet mash was supplied daily 48 h before the start of treatment and throughout treatment.

All the cell lines were grown in mouse cell line media as explained before[16,51]. At the commencement of the experiment mice received $1 \times 10^7$ cells injected intraperitoneally (IP) in 300 µL PBS. Treatment was usually started in HGS2 model at 6 to 7 weeks and at 9 weeks in 30200 mouse model. For survival experiments, mice were assessed daily and weighed twice weekly. Assessment of survival endpoint was based on 'moderate' severity as per the project license. The survival endpoints for mice were defined as a change in general health, specifically 15% body weight loss over 72 h or 20% over any time period, or inability to ambulate, or hunched posture, or difficulty breathing or signs of hypothermia as well as signs of ascites or palpable tumours exceeding an estimated size of 1 cm in diameter. Mice were specifically monitored for signs of autoimmunity such as the development of skin lesions, diarrhoea, weight loss and musculoskeletal pain manifesting as abnormal posture, reduced activity and abnormal mobility. The mice did not develop any of these symptoms. Their body weight was monitored two or three times weekly from the start of treatment.

Assessment of mice was made twice weekly by the same individual to limit inter-observer variability. Furthermore, in the majority of cases survival determinations were made by a trained animal technician who was not directly involved in the experimental design. Treatments were given by experienced animal technicians who were anonymous for the compounds/placebos given. For the time point experiment, mice were culled at a predefined time point (usually 1 weeks after the 3rd or the 4th dose of treatment, mid-treatment) for tissue collection to study the TME modulation as a result of the drug.

## Compounds used for in vivo experiments

**Carboplatin and Paclitaxel.** Carboplatin (Hospira) (10 mg/ml) and paclitaxel (Hospira) (6 mg/ml) were obtained from the pharmacy at St. Bartholomew's Hospital, London, were dissolved in 0.9% NaCl, and were administered to mice IP in 200 µL volume. Mice were treated with a combination of carboplatin 20 mg/kg + paclitaxel 10 mg/kg, based on average weight of all mice in the experiment. Control-treated mice received 0.9% NaCl 200ul once weekly IP.

**Foxp3 anti-sense oligonucleotide (Foxp3-ASO).** The murine surrogate of AZD8701[36] was obtained from AstraZeneca under MTA. The drug is an antisense oligonucleotide (ASO) that selectively inhibits FOXP3 transcription factor. Foxp3-ASO was used at a dose of 50 mg/kg administered twice weekly subcutaneously (sc). A control ASO (CTL-ASO) was given sc at the same dose and same treatment schedule for control mice.

**Anti-stabilin-1 antibody.** Monoclonal murine anti-stabilin1 receptor antibody (clone 1.26) was used for in vivo mouse treatment and was given at a dose of 100 µg per dose twice weekly IP for 6 weeks[43]. An isotype IgG1κ control antibody was given to the control group at the same dose and treatment schedule.

Mice in the experiments in Figs. 6C and S7C, D received in addition to the above controls inert vehicle (hydroxymethyl cellulose 0.5% in distilled water) by oral gavage, 100ul/day.

**Sample preparation for single cell RNA sequencing for human samples.** Human omental biopsy specimens were collected fresh directly from the operating theatre and processed immediately. Miltenyi biotec® tumour dissociation kit (Cat. number: 130-095-929) was used according to manufacturer instructions. Briefly, 2–3 grams of tissue were cut into 2–3 mm pieces using a scalpel then incubated in 4.7 ml of DMEMF12 and the kit enzyme mix at 37 °C for 20 min with mechanical dissociation using gentleMACS™ Dissociator in gentleMACS™ C- tubes (Cat. number: 130-093-237). The cell suspension was then filtered through 70 µm cell strainer followed by RBC lysis (BD Bioscience, Cat. number: 00-4300-54). Enrichment of live cells was done by magnetic sorting with magnetic labelled Annexin V microbeads (Miltenyi biotec, Cat. number: 130-090-101). Then the viable cell suspension was enriched for CD45+ cells using CD45 Microbeads (Miltenyi biotec, CD45 (TIL) microbeads, Cat. number: 130-118-780). The cells were then tested for viability and number of cells using automated cell counter. The cell suspension was immediately washed with PBS containing 0.04% BSA and no EDTA and suspended in the same buffer at a concentration of 1200 cells/µl. The viability and concentration of the cell suspension were above 90% (except for one sample had viability of 78%) and purity > 90%. Cells were tested by flow cytometry for purity.

**Sample preparation for single cell RNA sequencing for mouse samples.** Mice were culled by cervical dislocation. A laparotomy was performed, and the mouse omentum was dissected from surrounding structures and placed in PBS at 4 °C. Omenta were weighed using a AM100 analytical balance (Mettler). The sample was cut into pieces approximately 1 mm² in 2.5 ml of DMEMF12 supplemented with collagenase from Clostridium histolyticum (Sigma, cat. no.C9263) and DNAase I from bovine pancreas for 20 min with mechanical dissociation using gentleMACS™ Dissociator in gentleMACS™ C- tubes (Cat. number: 130-093-237). The resulting digest was then passed through a 70 µm cell strainer and flushed with fresh medium. Enrichment of live cells was done by magnetic sorting with magnetic labelled Annexin V microbeads (Miltenyi biotec, Cat. number: 130-090-101). The resulting cell suspension was centrifuged at 300 g for 3 min. Cells were then resuspended in PBS containing 0.04% BSA and no EDTA as above. The viability and concentration of the cell suspension were above 85%. Cells were tested by flow cytometry.

**Bulk RNA sequencing sample preparation, libraries and sequencing.** Mouse omental tumours were collected fresh and immediately placed in RNAlater™ buffer (Thermofisher, # AM7020) and stored at 4 °C till RNA extraction was done within 24 h using Qiagen RNAeasy kit (Qiagen, # 74004) according to manufacturer instructions. RNA quantification was done using nanodrop and Qubit. RNA quality was assessed using RNA integrity measured with Agilent 2100. RIN values ranged between 6.5–9 (except for 3 samples with RIN value 5.7 and 6, 6.3). Libraries were prepared with ribosomal depletion. Samples were

pooled and sequenced on Novaseq 6000 to depth of 40 M reads, 150 bp paired-end sequencing per sample.

**Flow cytometry.** Human omental tumours were dissociated into a cell suspension as above. The cell suspension was incubated with Fc blocking antibody at dilution 1:100 (Miltenyi biotec, Cat. number: 130-059-901) and fixable viability dye (eBioscience, Cat. number:65-0866-14) at dilution 1:1000 in 50 µl of PBS for 15 min at 4 °C in the dark then washed twice with PBS containing 2.5% BSA and 2 mmol/L EDTA (FACS buffer). Cells were then stained with flow cytometry antibodies in FACS buffer for 20 min at 4 °C. Cells were then washed twice with PBS containing 2.5% BSA and 2 mmol/L EDTA then fixed in 2% formalin saline then washed and suspended in 200 µl of PBS containing 2.5% BSA and 2 mmol/L EDTA. Appropriate florescence minus one (FMO) controls were used. Flow cytometry data were acquired on LSR fortessa cell analyser (BD Bioscience). Data analysis was done in FlowJo Version 10.7.3 (Treestar inc).

For intracellular staining (eBioscience™ Foxp3 / Transcription Factor Staining, cat number 00-5523-00) kit was used according to manufacturer's instruction. Briefly, cells were stained as above for the extracellular staining then fixed and permeabilized using (eBioscience™ Foxp3 / Transcription Factor Staining, cat number 00-5523-00) for 1 h then washed once and the intracellular antibodies were then added diluted in permeabilization buffer provided in the kit.

**Human monocyte isolation and macrophage culture.** Human monocytes were isolated from leukocyte cones from the NHS Blood and Transplant service. PBMC were isolated using Ficoll-Paque PLUS (GE Healthcare, cat. no. 17-1440-03 AG). CD14+ monocytes were isolated using CD14 microbeads (Miltenyi, cat. no. 130-050-201) as per manufacturer's protocol. CD14 monocytes were plated in a 48 well plate 1 million cell per well. Monocytes were cultured in DMEM-F12 medium supplemented with 10% FBS, pencillin/streptomycin + L-Glutamine and supplemented with 100 ng/ml of MCSF for 5 days. Cells were treated on day 5 with either isotype control IgG (10µg/ml) or anti-stabilin1 antibody (10µg/ml) and on the 6th day, polarization was done by IL-4 (20 ng/ml), IFNγ (20 ng/ml) or dexamethasone (100 µg/ml), carboplatin (5 µg/ml) or paclitaxel (18 ng/ml) added for 24 h. On day 7, Cells were then used for subsequent assays such as imaging, phagocytosis or harvested for flow cytometry.

**Human monocyte derived dendritic cell culture.** CD14+ monocytes isolated from leucocyte cones from healthy donors as explained above were cultured in 48 well plate as 1 million cell per well. Monocytes were cultured in DMEM-F12 medium supplemented with 10% FBS, pencillin/streptomycin + L-Glutamine and supplemented with 100 ng/ml of GMCSF and 20 ng/ml of IL-4 for 4 days then on day 5 20 ng/ml of IFNγ is added for dendritic cell maturation[32,33].

**Human T cell isolation and activation.** Human PBMC were isolated from leucocyte cones from the NHS Blood and Transplant service using Ficol Plaque PLUS (GE Healthcare, cat. no. 17-1440-03 AG). Un-labelled CD3 T cells were magnetically separated with pan T cell isolation kit (Miltenyi biotec, Cat. number: 130-096-535) according to manufacturer's protocol. Un-labelled T cells were isolated by depletion of non-T cells which are retained in the magnetic column and the un-labelled T cells are collected in the flow through (i.e negative selection).

T cell activation was carried out using TransAct® (Miltenyi biotec, Cat. number: 130-111-160) as per manufacturer's instruction. One million CD3 T cells were plated per well in 48 well plate in DMEM F12 + 10% FBS+ pencillin/streptomycin + L-Glutamine. 10ul/ml of TransAct were added and 20 IU/ml of recombinant human IL-2 and incubated at 37 °C, 95% humidity and 5% CO2 for 3 days. T cells were generally harvested on Day 4 for flow cytometry or T cell killing assay. Tumour supernatant (SNT) or macrophage conditioned medium or FOXP3-ASO (5µg/ml) were added at time of plating the T cells. FOXP3-ASO was replaced every 4 days.

**T cell killing assay.** This technique was developed by Kitamura et al.[52]. Briefly, the aim is to test the effect of macrophages on T-cell cytotoxicity against the tumour cells. The tumour cells were labelled with red fluorescent mKate2 protein that is produced after lentivirus transduction of the cancer cells. T-cells were isolated from human leukocyte cones and activated using TransAct bead and recombinant human IL2. T cells were cultured in macrophage-conditioned medium for 3 days. T cells were then incubated with the target cells (LV labelled red cancer cells) at ratio 4:1 in the presence of CYTOTOX® (Essence bioscience, Cat # 4632) green dye at concentration 1/5000. Dead cells will acquire the green CYTOTOX dye. Cells were imaged in Incucyte S3 for 48–72 h with images captured every 2 h in phase, red and green (example of those images is included in Supplementary Fig. S4C). The dead cancer cells will be separated from the dead T cells by size criteria and overlapping with the red colour. The number of dead cancer cells was plotted over time to generate the curves shown in Fig. 4D, J. The values were then exported from the Incucyte software and used in PRISM to generate an area under the curve value for each individual replicate of each treatment condition. Those values were plotted to produce Fig. 4D,E and statistical significance was then studied using unpaired *t* test.

**Phagocytosis assay.** The pHrodo® Red cell labelling kit from (Essenbio® Sartorius Cat # 4649) was used in this assay. The idea of the assay is that the target cells are labelled with the pHrodo® dye that is uncoloured at normal pH (7–9), however, it becomes red in acidic PH (4–5.5). If the cells are ingested, the pH in the phagolysosome is acidic and the ingested cells will appear bright red. Optimization of the pHrodo® dye concentrationto label the target cells, was done as an initial step as per manufacturer instructions and a concentration of 500 ng/ml was used to label AOCS1 cells. The plate was imaged every 30 min in Incucyte S3 using phase and red. In each condition three biological replicates were included. The phagocytic capacity was measured by measuring the red object mean intensity and plotted on a curve along time. Values were then exported from the Incucyte software and imported into PRISM. Area under the curve (AUC) was generated for each replicate in each experimental condition and then used to compare experimental conditions. Student t-test was used to study the significance of the treatment on phagocytosis at different treatment conditions.

**DQ™-Ovalbumin assay.** DQ™-Ovalbumin (Invitrogen, Cat# D12053) is a self-quenched conjugate of ovalbumin that exhibits bright green fluorescence upon proteolytic degradation. This substrate is specifically labelled with BODIPY® FL dye which is pH insensitive. The lyophilised powder vial (1 mg) is diluted in 1 ml of PBS as per manufacturer instructions to constitute a solution of ovalbumin of 1 mg/ml. 50ug/ml of ovalbumin was added per well on macrophages matured under different differentiation and treatment conditions mentioned above, three replicates (3 donors) for each condition. The plate was then imaged every 30 min in Incucyte S3 using phase and green. The DQ™-Ovalbumin was readily ingested by macrophages and emits bright green fluorescence on proteolysis. The number of green fluorescence objects (corresponding to degraded antigen/ovalbumin) increased over time. Values were then exported from the Incucyte software and imported into PRISM. Area under the curve (AUC) was generated for each replicate in each experimental condition and then used to compare experimental conditions. Student t-test was used to study the significance of the effect of treatment/cytokine on antigen degradation.

**LysoSensor™ green assay.** LysoSensor Green DND-189 (Invitrogen, Cat# L7535) becomes more fluorescent in acidic environment and is PH sensitive so can be used to measure the acidity of acidic organelles. 1 μM of dye was added per well containing either macrophages or dendritic cells under different treatment/polarization conditions in tri-replicates as explained above. Then the plate was imaged in incucyte S3 every 30 min. The green fluorescence intensity of the whole well increases over time as more LysoSensor is ingested in acidic lysosomes in the cell. Values were then exported from the Incucyte software and imported into PRISM. Area under the curve (AUC) was generated for each replicate in each experimental condition and then used to compare experimental conditions. Student t-test was used to study the significance of the effect of treatment/cytokine on the acidity of the lysosomes.

**Protein Array analysis.** Human Cytokine Proteome profiler Array kit (biotechne, #ARY005B) was used to study the cytokine profile of FOXP3-ASO treated and CTL-ASO treated T cells in vitro. Supernatant was collected from T cell culture on day4 and then used fresh according to manufacturer instructions. The membranes were imaged using Amersham Imager 600 (GE Healthcare). Images were imported into ImageJ software and Protein Array Analyzer plugin was used to quantify the dots on the array membrane.

**Immunohistochemistry (IHC) protocols.** Formalin-fixed paraffin-embedded human or murine tissue sections were subjected to immunohistochemistry. Briefly, after deparaffinization and rehydration, antigen retrieval was done using antigen retriever with PH 9 then the sections were subsequently washed, treated with 3% $H_2O_2$ (Fisher Scientific, H/1800/15) in methanol for 10 min followed by another washing step with PBS. After blocking with BSA 5% in PBS for 1 h at room temperature, sections were incubated with the primary antibody for either 1 h at room temperature or overnight at 4 C depending on the antibody. Sections were then appropriately washed and incubated with the appropriate secondary antibodies followed by washing. Colour was developed with Diaminobenzidine substrate-chromogen (Dako Liquid DAB+ Substrate Chromogen System, K3468 Dako) or Alkaline phosphatase red substrate (Vector Vector® Red Substrate Kit, SK-5100) and tissues were counterstained with Gill's hematoxylin I (Sigma-Aldrich, GHS1128), washed, dehydrated in ethanol and xylene and mounted in DPX (Sigma-Aldrich, 06522). The full list of primary and secondary antibodies are provided in the Reporting summary. Immunohistochemistry sections were imaged using the Panoramic digital slide scanner (3DHISTECH) or NanoZoomer. QuPath version 0.4 digital image analysis was used to quantify staining using positive cell detection per mm².

### Treatment used in the in vitro experiments

**Carboplatin and paclitaxel.** Carboplatin (Hospira) (10 mg/ml) was used at a dose of 5–10 μg/ml for in vitro experiments. This was the maximum dose that was not toxic to the macrophages. This dose agreed with the doses used by Dijkgraaf et al.[53]. Paclitaxel (Hospira) (6 mg/ml) was used at a dose of 18 ng/ml for in vitro experiments. Similarly, this was the maximum dose that was not toxic for macrophages and was consistent with the in vivo dose used for HGSOC patients[54,55].

**FOXP3-ASO.** AZD8701 was used at dose of 5 μg/ml[36]. The drug was replenished every 4 days in the culture.

**Stabilin-1 receptor antibody.** Monoclonal anti-stabilin1 receptor anti-mouse antibody (clone 1.26) was used for in vitro experiments[42,43]. It was initially obtained as gift from Professor Julia Kzhyshkowska, Heidelberg University, Germany then for subsequent experiments it was obtained from InVivo Biotech company under MTA with Prof Kai Schledzewski, Heidelburg, Germany. A dose of 10 μg/ml was used for in vitro cultures[56]. Rat Monoclonal human anti-stabilin-1 receptor antibody (clone 9–11) was used for in vitro human experiments at a dose of 10 μg/ml. The antibody was obtained with kind permission from Professor Maija Hellman, university of Tuku, Finland and purchased from InVivo Biotec company.

### Computational methods

**Sequencing and Raw data processing of single cell RNA seq.** The aim was to capture 5000–10,000 cells per sample. 30,000 cells were loaded on the 10X chromium controller chip as per 10X protocol. Gel beads in emulsion (GEM) were then used for reverse transcription to generate cDNA. The quality of the cDNA was checked with bioanalyzer before library preparation. Libraries were prepared using 5'prime kit. Each sample was sequenced to a depth of approximately 500 million reads (i.e., around 50,000 reads per cell depending on the number of cells in each sample). The samples were pooled at equal amount in one single library and sequenced on Illumina Novaseq600 using paired end 150 bp. Data pre-processing was done using Cell Ranger 6.0.2 Single-Cell Software Suite (https://support.10xgenomics.com/). Analysis was done using cellranger multi pipeline to get consistent cell calling between the V(D)J and the gene expression data. GRCh38-2020-A and mm10-2020-A transcriptomes were used as reference for the gene expression libraries, vdj_GRCh38_alts_ensembl-5.0.0 and vdj_GRCm38_alts_ensembl-5.0.0 were used as reference genomes for the VDJ libraries. Cellranger did the assembly, alignment, annotation and quantification of the reads. The output of cellranger multi includes 3 folders, feature counts, vdj_T and vdj_B. The feature count files were used as an input to Seurat package[57] for analysis of gene expression data. The filtered contig annotation csv files for TCR and BCR were used as an input for scRepertoire package (version 1.3.3)[58], each separately, to perform the clonotype analysis, CDR3 distribution and repertoire overlap and diversity.

**Analysis of data of single-cell RNA sequencing, Clustering and visualisation and differential expression.** For each sample (or library), filtered feature counts were imported in R (version 4.0.2) and converted into Seurat object using Seurat version (4.0.5) using *CreateSeuratObject* function. Low quality cells were filtered out based on the percentage of mitochondrial genes and number or genes. The percentage of mitochondrial genes and ribosomal genes per cell were calculated. Cells with high mitochondrial genes (>25%) or less than 200 genes were excluded for low quality. Cell cycle scoring was calculated using *cellcyclescoring* function of Seurat. TCR, BCR, mitochondrial genes were removed from the matrix before analysis. Highly variable genes were calculated using SCTransform package of Seurat regressing out the effect of mitochondrial genes, ribosomal genes and cell cycle scoring difference. Principal component analysis (PCA) was done using the first 50 PCA for clustering at resolution of 1. Seurat objects from all individual samples were merged together and clustered, then the clusters- were manually annotated based on canonical markers for the major cell populations, T/NK cells, B cells, myeloid cells and non-immune cells. Clusters from each major population were subsetted then integrated and batch corrected using Seurat reciprocal PCA with k.anchors = 5. After integration, the integration transformed expression matrix was used for dimensionality reduction and clustering. Principal components analysis (PCA) was performed using first 50 principal components for Louvain clustering of cells at a resolution parameter of 0.2–1.8. Uniform manifold approximation and projection (UMAP) was performed on the same PCs for visualisation. The RNA normalized expression matrix was used for all differential expression and gene set level analyses. To identify cell subtypes and states within each major batch corrected cell population, differentially expressed genes between clusters were calculated using *FindAllMarkers* function with default parameters.

To study the differentially expressed genes between different treatment conditions, Seurat *FindMarkers* function was used (i.e, postchemotherapy versus prechemotherapy) using RNA assay. In the mouse dataset, the effect of ambient RNA was regressed using Celda package[59] with *decontX* function run on raw counts and the decontaminated count matrix was then used for further analysis.

**Trajectory analysis.** The cell differentiation lineages were predicted from the UMAP embeddings and the PCs from Seurat using the slingshot package[17].

**Similarity analysis of clusters from human and mouse datasets.** To study the similarity between human and mouse clusters, we performed a systematic similarity analysis comparing clusters of each major cellular population from mouse dataset (test dataset) to the human dataset (training dataset). For this purpose, we used a method that was published by Zhang et al.[37]. The code was kindly shared by Professor Ziyi Li on request. A brief explanation of the principals of this method is as follows: down-sampling of each cluster to the minimum size of all clusters in the training dataset was done to eliminate possible bias due to different sizes of clusters, with a minimum of 50 cells per cluster. A logistic regression model was trained using elastic net regularization[60] and used cv.glmnet function from the glmnet package (version 2.0.16) to fit a series of n binomial logistic regression models with parameters alpha = 0.99, where n is the number of clusters in the training data. A 10-fold cross validation was performed in each case. These models were used to calculate a predicted logit of each cell in the test data for each cluster from the training data with an offset of 0. Predicted logits were then averaged within each cluster and converted to probabilities for visualization, which indicated the similarity of clusters from the test data to those from the training data.

**TCR and BCR analysis.** The filtered contig annotation csv file for TCR and BCR were used as an input for scRepertoire package (version 1.3.3 and 1.7)[58], each separately, to perform the clonotype analysis, CDR3 distribution and repertoire overlap and diversity. Startrac package[61] was used to calculate the Expansion index and transition index between clusters.

**Pathway enrichment analysis.** Gene set variation analysis from the GSVA R package (version 1.40.0) was used for gene set enrichment analysis. The gene sets were exported using the GSEABase package (version 1.54.0) from MSigDB for human and mouse. GSVA generates an enrichment score for each pathway in each cell. A mean score from all the cells in the cluster was calculated and used for plotting purposes. Limma package (version 3.52.4) was used to calculate the significance of the differences in pathway activities between two treatment conditions (i.e PDS and NACT). Fgsea was used to study the significance of enriched pathway between different treatment conditions in bulk RNA seq data. The preranked gene list was impirted directly from DESEq2[62] object into fgsea.

**Analysis of bulk RNA sequencing.** Raw sequencing data from the Illumina platform were converted into Fastq files and aligned to the reference genome mm10 using Salmon. Counts were then imported into R using tximport[63] to create raw matrix counts. DESEq2[62] was used for data processing, normalization, and differential expression analysis according to standard procedures. Bulk RNA sequencing for human samples were analysed using EdgeR[64].

### Quantification and statistical analysis

Graphic representation of data and statistical analysis was performed in Prism Version 9.0. Data were tested for normality using the Shapiro test. If the data were normally distributed, an unpaired Student's t-test with Welch correction was used for analysis of differences between two groups. For unpaired non-parametric data, Mann–Whitney test was used. For paired samples Wilcoxon matched-pairs sign ranked test was used. For multivariate data analysis, one or two-way analysis of variance (ANOVA) was used for assessment of group differences with Tukey's post-test applied. The measurements of all statistical values were performed using GraphPad Prism 9.0. Error bars indicate median and 95% confidence interval. Experimental replicates were always included in the figure legends.

Spearman correlation and regression analysis were used to assess the relationship of STAB1 and FOXP3 levels (quantified on IHC) with survival and PFS. Kaplan–Meier survival curves for human data were plotted in R using survminer, survival and ggpubr R packages, and the log-rank (Mantel–Cox) test was used to compare survival curves. Kaplan–Meier curves for the mouse experiments were plotted in PRISM version 9.0.

### Reporting summary

Further information on research design is available in the Nature Portfolio Reporting Summary linked to this article.

## Data availability

Human single-cell RNA-seq data have been deposited at GEO database (Accession number: GSE224392). Mouse single-cell RNA-seq data have been deposited at GEO database (Accession number: GSE224389). Bulk RNA-seq data have been deposited at GEO database (Accession number: GSE224091). All datasets are publicly available. The analysed data is provided in the supplementary tables and figures.

## Code availability

No custom codes have been used to analyse the data and the paper did not produce new codes.

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

## Acknowledgements

We would like to thank the Genomics core facility in the Blizard Institute, Queen Mary University of London, especially Dr Charles Mein and Ms Eva Wozniak, for running the 10X chromium for the single cell experiments. We thank Dr Gordon Beattie, University College London for advice on scRNA-seq data analysis. We would like to thank Dr Ziyi Li, Shanghai Institute of Immunology, Shanghai Jiao Tong University for sharing the code for the similarity analysis from their previous Cell paper (ref. 29). We thank Professor Siamon Gordon, University of Oxford for his advice on the anti-stab1 antibody. We thank Dr Kai Schledzewski, Heidelberg University, Germany, for providing permission to use the anti-stab1 (clone 1.26) for in vivo experiments which was purchased from in Vivo Biotech. We thank Professor Maija Hollmen for provide permission to use human anti-stab1 antibody (clone 9.11) for in vitro work purchased from in Vivo Biotech. We thank Dr Christina Schmuttermaier for providing advice on staining protocols for anti-stab1 antibody in human and mouse. We would like to thank the BSU and animal technician services in Barts Cancer Institute (BCI) who helped with animal experiments which were partially funded by Cancer Research UK Major Centre Award A18066. Services from the flow cytometry facility and microscopy core facility were funded by Cancer Research UK Major Centre Award A18066. We would like to thank the surgical teams at Barts Trust and St George's University Hospitals trust who facilitated obtaining human samples from theatre. This work is funded by Wellbeing of Women scholarship grant ELS906 and clinical research training fellowship grant RTF1013 (S.E.), CRUK programme grant A25714 (C.B., E.M., F.B.), CRUK core grant award A18066 (E.M., J.W.), Barts Charity grant ECMG1B6R (R.M.), UKRI Frontier Research Grant EP/X028704/1 (F.B.).

## Author contributions

S.E. conceived the project, secured the funding planned, conducted the experiments, data analysis, the bioinformatics analysis and wrote the manuscript. C.B. provided advice with the mouse experiments and previously developed the mouse models. E.M. provided bioinformatics and statistical advice for data analysis. J.W. provided advice for single-cell RNA sequencing data analysis. L.C. and S.B. provided AZD8701 compound and advice on the dose and scheduling for the in vitro and in vivo experiments. R.M. is the Principal Investigator of the SIGNPOST study which provided ethics for human sample collection and led the surgical team that provided the samples. J.K. provided the anti-stab1 antibodies used in one of the in vivo studies and for IHC staining, provided advice on stabilin-1 staining protocol and anti-stab1 antibody 1.26 doses used in the in vitro experiment for stabilin-1. F.B. supervised the work and wrote the manuscript. All other authors reviewed and edited the manuscript.

## Competing interests

S.E., C.B., E.M., J.W. and J.K. declare no conflict of interest. L.C. and S.B. are paid employees of AstraZeneca. R.M. declares honorarium for advisory board membership from Astrazeneca, MSD and EGL. F.B. is on the Scientific Advisory Board of iOmx Therapeutics AG and has received honoraria from GSK.
