## [Transparent Peer Review file · Nature Communications]

Immunotherapy that improves response to chemotherapy in high-grade serous ovarian cancer

Corresponding Author: Dr Samar Elorbany

Version 0:

Reviewer comments:

Reviewer #1

(Remarks to the Author)

The manuscript by Samar Elorbany et al. titled 'Immunotherapy that improves response to chemotherapy in HGSOc' is an elegant study that identifies immune cell populations in patients and mice (mainly omental metastases) before and after chemotherapy using scRNA-seq. Honestly, I was quite surprised that the cell populations could be identified using scRNA-seq and 7 total patients, but the data appears solid and impactful. More importantly, the authors identify 2 targets and their mechanisms of action, STAB1 and FOXP3, in human cells in vitro and show the therapeutic effect of targeting these two proteins on HGSOc using in two mouse models.

Major Findings:

Note, the manuscript is quite lengthy and making more concise would increase clarity of manuscript.

Validation of the findings in many patient samples by the authors is extremely valuable. However, the study would be greatly enhanced with some spatial context of these populations in human and mouse tumors. The authors should consider including IHC images or performing multiplex IMF on the 7 patients included in original study. Where are these immune cells located? In relationship to each other? To cancer cells? to stroma? Furthermore, while scRNA-seq allowed the authors to identify immune cell populations and gene expression in these immune cells after chemotherapy, other tools are available to define these immune cells including flow cytometry/mass cytometry etc. with greater specificity.

In addition, the figures include a lot of pertinent data but are illegible unless zoom in quite a bit.

The immune profiling of the mouse models will be extremely valuable as a tool for future researchers so the authors should consider sharing all data raw etc. for others to use to find ideal models to test future therapeutics that activate/exhaust cause infiltration etc. of the immune system

Reviewer #2

(Remarks to the Author)

What are the noteworthy results?

The authors have taken on several objectives. They performed single cell RNA seq on cells from omental metastases (patient-unmatched) from seven HGSOc patients.

They studied the differentially enriched macrophage clusters. They identify 2 clusters (an antigen presenting cluster) and an ECM-regulator and angiogenesis cluster.

Following neoadjuvant chemotherapy they identified upregulation of multiple macrophage scavenger receptors (STAB1, MSR1, CD163) and down regulation of ECM-regulation genes.

They found that STAB1 upregulation was associated with receipt of NACT (controlled by review of sample from patients with a primary debulking surgery), and that STAB1 upregulation was associated with worse PFS. They hypothesized that blocking STAB1 could improve antigen presentation after NACT and could repolarize macrophages towards an APC phenotype.

The authors then report the result of an anti-stab1 antibody on macrophages using monocytes from healthy donors. They also found that T cell cultured in conditioned medium from anti-stab1 antibody treated macrophages showed improved

killing.

They identified five Treg clusters defined by FOXP3 expression, and they found that NACT significantly upregulated FOXP3. This finding was confirmed using 42 patient matched pre/post- chemotherapy samples. High FOXP3 post-chemo was associated with worse PFS.

They report that T cells treated with AZD8701, a FOXP3 anti-sense oligonucleotide demonstrated improved killing and reduced T cell exhaustion.

They also analyzed cells from a syngeneic HGSOc mouse model. They validated the finding of increased STAB1 expression in omental tumor in mice treated with chemotherapy.

They validated their results by testing AZD8701 and an anti-stab1 antibody (as monotherapy and in combination) with chemotherapy in two HGSOc mouse models. They found a marked improvement in progression free survival in the syngeneic mouse models with the triple combination (AZD8701 + anti-stab1 antibody + chemotherapy).

They submitted the treated tumors from the mouse models for bulk RNA sequencing. They found a significant upregulation in antigen presentation pathways and TCR signaling, as well as allograft rejection in the anti-stab1 antibody treated mice. Treatment with AZD8701 resulted in higher levels of immune activation pathways.

Will the work be of significance to the field and related fields? How does it compare to the established literature? If the work is not original, please provide relevant references.

Yes, this work will be of significance to the field. Chemotherapy response of HGSOc to first-line therapy with carboplatin and paclitaxel remains suboptimal, and most patients will go on to have incurable, recurrent disease. They present intriguing preclinical findings outlining potential interactions between response to chemotherapy and multiple points of immune regulation. They have shown that both FOXP3 and STAB1 are relevant immune regulators following NACT. They further demonstrate the FOXP3 and STAB1 are potential therapeutic targets in HGSOc, and they present compelling preliminary preclinical data using syngeneic mouse models.

Does the work support the conclusions and claims, or is additional evidence needed?

Yes, their work supports these claims. It is noted that the identification of these targets is from observational data, rather than a prespecified hypothesis identifying FOXP3 and STAB1 as critical regulators in the HGSOc tumor microenvironment. Nonetheless, the authors identify a potential new therapeutic approach for the treatment of HGSOc. Further clinical study will be needed.

Are there any flaws in the data analysis, interpretation and conclusions? Do these prohibit publication or require revision? Lines 568-569 it is written "There is hardly any information on pre-clinical models for relapsed HGSOc." I am not sure what is meant by this. There are many publications reporting the results of HGSOc models (ex. Cook, D.P., Galpin, K.J.C., Rodriguez, G.M. et al. Comparative analysis of syngeneic mouse models of high-grade serous ovarian cancer. *Commun Biol* 6, 1152 (2023). <https://doi.org/10.1038/s42003-023-05529-z>). Could the authors please clarify this point?

The discussion would be further enhanced if mentioned were made of the lack of response of HGSOc to anti-PD1 therapy.

Is the methodology sound? Does the work meet the expected standards in your field?

The methodology is clearly outlined and appears to be sound.

Is there enough detail provided in the methods for the work to be reproduced?

I do not perform these techniques personally, and I will defer to the reviewers with more expertise in these methods. The methods appear generally sound and logically presented.

Reviewer #3

(Remarks to the Author)

Elorbany et al report scRNA-seq on immune enriched cells from omental metastasis from seven patients; three patients had primary debulking surgery and four patients received NACT. They also present mechanistic studies focus on myeloid and T-reg activation on PBMCs, mouse models investigating two immunotherapy treatments in ovarian cancer. The main findings are focused on targeting FOXP3 and STAB1 to enhance the effect of chemotherapy in mouse models.

The main concern is that the analyses of the human samples were done in patient-unpaired samples, introducing an inherent bias towards different tumor characteristics due to patient selection. Therefore, the data cannot be used to support causal claims of NACT effect (e.g. "NACT induced X"). This concerns all of the results that are described on the immune cell subpopulations from the human samples (pages 5-10). The NACT-induced changes in cell phenotypes should be analysed in patient-matched samples, or alternatively the wordings changed throughout the manuscript. The use of patient-unmatched samples also does limit the conclusions that can be made from the study.

Another major concern is the methodological validity of the scRNAseq analyses. Specifically, some of the filtering cut-offs are not justified, and the differential gene expression analysis presents unusually low cutoff values ($\log_2FC=0.25$). Typically minimum values of 0.5 or 1 are used to capture the meaningful effect sizes. Similarly the justifications for p-value cut-offs and multiple hypothesis adjustment are missing. A different set of cutoffs both for the adjusted p-value and \log_2FC are applied in the bulk-RNA seq validation. The statistical analysis needs to be revisited and better explained to demonstrate the validity of the analyses.

In general, the manuscript logic (analyses and figures) and the results are quite hard to follow. The manuscript is partially well but sometimes poorly written as if it was compiled from text written by different persons. The writing should be

harmonised. The manuscript would benefit from clearly focusing on describing the scRNAseq and functional findings of only the most relevant cell populations (e.g. the macrophages and the phagocytosis part is intriguing).

Specific comments:

Fig 1 I (and later similar correlations): PFS correlations should not be made with linear regression - only Kaplan-Meier analyses should be shown. Anyhow R value of 0.3 barely indicates a significant linear relationship of the variables.

Figures 2 and 3 are largely descriptive- focusing on describing the clusters. The text is difficult to follow, the logic of describing the results and the writing should be improved. The main conclusions from this part of the manuscript remain unclear. The authors should rewrite these parts and focus on answering specific scientific questions. It is unclear why the authors focus on STAB1?

It would be interesting to specify the molecular profile of the patients in which scRNA-seq has been performed, since this cannot be found in either the text or the figures.

The STAB1 PBMC analyses are interesting. What was the rationale to investigate FOXP3 targeted therapy? What was the biological rationale to combine STAB1 and FOXP3 targeted agents?"

From the in vivo work, it is difficult to conclude what are the main findings. Also - the differences in the Kaplan-Meier analyses are not really dramatic, and I wonder how relevant are these differences? Figure 6A-C - it seems that most of the immunotherapy benefit was driven by the FOXP3 - ASO? No significant benefit was gained via adding STAB1? What was the rationale of combining these treatments?

The authors conclude that the anti-stab1 treatment changed the way macrophages process the phagocytosed antigen, potentially strengthening the antitumor immune response. I wonder if there could be a deeper mechanistic characterisation, at the gene expression level, addressing how is the antigen processing different in pre and postNACT settings. In the bulk RNA-seq results from the in vivo experiment tumours they do not go into the differential expression analysis but only to the pathway-level information.

In terms of logic, the discussion needs a lot of work. In stead of discussing the relevance of the findings in a broader context, the authors only discuss the justification of their study and limitations. There are several overstatements in the discussion (for example "In this paper, we have shown the efficacy of our identified targets in treating relapsed HGSOc").* None of the 12 ovarian cancer patients with relapsed disease responded to an Anti-Cleaver-1 antibody treatment (<https://doi.org/10.1016/j.xcrm.2023.101307>)

Minor comments

- In Fig1A text is hardly visible and it is challenging for the reader to distinguish between clusters. I would suggest increasing font size and changing the colorscale. In Fig. S1E I would add again the legend for the clusters. These comments also apply for Figs 1,2

- Fig 1D and S1C: annotations of threshold values in the axes are recommended. Colorscale is missing in Fig S1C. Also, the range of the colorscale is not informative in Fig 1D, since color differences are barely noticed. I recommend going for a discrete scale instead of continuous, (>0.05, 0.05-0.01, <0.01)

- Cutoff values are too small, fold-change is only 0.25. At least I would recommend a 1 log2 FC value to account for these differences.

- Lines 128-130: "Compared to PDS samples, we found significant upregulation of macrophage scavenger receptors, e.g., STAB1, MSR1, CD163, phagosome proteolytic cathepsins, e.g., CTSB, CTSL and CTSD and complement in NACT samples" In which figure are these results presented? Some of these genes are not presented in Fig1D.

- In Fig 1F, it looks like the expression level of SIGLEC1 was significantly higher in NACT than in PDS?

- In ICGC dataset: Are not all the ICGC dataset tumors PDS? What was the rationale for choosing this dataset?

- Lineage trajectory analyses - what was the scientific question and main result that were obtained from the analyses? It looks like in Fig 1A, for the UMAP the M0 population is found in two separate clusters. Same phenomenon is seen in M2. Also, in the trajectory analysis it seems that one of these clusters is involved in the maturation of M1 and M9 (belonging to the ECM-category), whilst the other seems to be part of lineages comprising APC phenotypes. This is unconventional and needs further clarification.

- The authors claim that (171): Each patient and both treatment groups had a variable percentage of the dendritic cell clusters with no difference between PDS and NACT. Yet authors show upregulation of AP and phagolysosome-related pathway enrichment in NACT

Reviewer #4

(Remarks to the Author)

The authors carried out single-cell RNA sequencing of tumor immune cells from high-grade serous ovarian cancer (HGSOc) samples and carried out validation studies. Please consider the following:

1) Page 5. This reviewer tried to locate Table S1 but could not find it. Additional clinical details on the patients would be important to present, including: 1) what exact chemotherapy drugs did patients receive on NACT? 2) Did any patients receive bevacizumab with their NACT? 3) Did any patients have germline or somatic BRCA mutations or HRD? 4) Did any

patients receive maintenance therapy?

2) To compare with human cancer samples, were they selected from patients with BRCA mutation or HRD? Was the histology from the two models used confirmed to be high grade serous?

3) For mouse samples, it would be helpful to see the cell selection criteria prior to scRNA seq.

4) The authors claim that the cell subpopulations changed after neoadjuvant chemotherapy. However, the cohort used has NACT and PDS samples and these were not matched for each patient. Thus, it is possible that some of the differences might be related to inter-patient heterogeneity rather than chemotherapy. At least some validation in matched samples should be performed.

Version 1:

Reviewer comments:

Reviewer #1

(Remarks to the Author)

The authors included key additional information in the manuscript and made the manuscript more concise. The mouse studies from analysis of immune cells to different treatments and validation of signatures is elegant and robust. However, important concerns on the acquiring/validation of this data from all reviewers needs to be addressed sufficiently in the manuscript.

Major

The authors use 42 patient matched samples to show levels of STAB in tumors but only use n=3 PDS and 4 NAPT patients in sc analysis. Further the authors showed that the overall #s of macrophages, etc. and overall expression level of STAB, etc. was different in pre- vs post-chemo patients (n=42). However, this data did not explore or confirm: "We concluded that NACT increased macrophage infiltration and phagocytic activity leading to 167 increased antigen presentation. However, as NACT also upregulated the macrophage 168 scavenger receptor, STAB1, this could lead to excess antigen degradation and less efficient 169 antigen presentation." Authors did not investigate or validate an increase in STAB expressing populations of macrophages after chemotherapy. As suggested in first review, the authors could address this by including patient-matched and more patients in sc analysis and show UMAPS of both sets and calculate differences M0-M8 populations etc., CD4 cell populations ; the authors could perform co-IMF or multiplex IMF etc to show increase in STAB producing macrophages, FOXP3 expressing CD4 cells; OR the authors could perform flow cytometry on digested tumors as shown in mouse study. The IHC shown and survival analysis are convincing to show that patients with higher tumor levels of STAB...but do not measure infiltration of macrophages and what type of macrophages esp. STAB expressing macrophages.

Reviewer #2

(Remarks to the Author)

As I had previously noted, chemotherapy response of HGSOc to first-line therapy with carboplatin and paclitaxel remains suboptimal, and most patients will go on to have incurable, recurrent disease. The authors present intriguing preclinical findings outlining potential interactions between response to chemotherapy and multiple points of immune regulation. They have shown that both FOXP3 and STAB1 are relevant immune regulators following NACT. They further demonstrate the FOXP3 and STAB1 are potential therapeutic targets in HGSOc, and they present compelling preliminary preclinical data using syngeneic mouse models.

I appreciate the authors response to my comments, as well as to the comments submitted by the other reviewers. My comments have been satisfactorily addressed. The revisions to the manuscript language have helped to improve clarity. The addition of previously published phase 1 clinical trial data helps to contextualize this finding in the clinical research landscape.

It is noted that this manuscript includes the results from multiple experiments using both patient-derived samples and syngeneic mouse models. It can at times be difficult to follow the progression of experiments, as has been noted by other reviewers. However, I do think the adequate information can be discerned from a careful review of the text and figures. I would encourage the authors to be as consistent as possible in their identification of each mode, both in the text and in the attached figures, to improve clarity.

Reviewer #3

(Remarks to the Author)

The authors have addressed most of my concerns. However, question one remains unanswered. While I share the authors' concern on the limitation of access to paired pre and post NACT patient samples, it is critical that the data supports the claims/conclusions made in the manuscript. These, unpaired data cannot be used to support causal claims of "NACT effect" (e.g. "NACT induced X"). This starts already in the first sentence of the results (row 105: To study the effect of chemotherapy), as one cannot really study the effect of chemotherapy in unpaired samples. The data would support conclusions that e.g. XX "was enriched in post NACT" samples, or that the study investigates the differences in samples before or after chemotherapy. I would ask the authors to tone down here their statements on the results due to this fundamental difference in the research setting and experimental design to avoid misconclusions. This concerns all of the

results that are described on the immune cell subpopulations from the human samples. This includes the abstract, results (eg Rows 166-167, 183, 193, 222, 239, 260, 265). Discussion rows 530,

Specific comments.

Comment 1. It would be interesting to specify the molecular profile of the patients in which scRNA-seq has been performed, since this cannot be found in either the text or the figures.

Response: We did not conduct bulk RNAseq on the scRNAseq samples used for this paper, therefore it is not possible for us to carry out any molecular classification.

-> Here I mean BRCA mutation/HRD. Is this information available?

For Specific comments 4-7. In the response letter, the authors don't specify what corrections they made to the manuscript. It would be easier to evaluate how the manuscript has been improved if the changes to the manuscript were annotated in the response letter.

The last paragraph of discussion is nice, but would need a language check.

Reviewer #4

(Remarks to the Author)

Comments have been addressed.

Version 2:

Reviewer comments:

Reviewer #1

(Remarks to the Author)

I appreciate the authors response to my comments, as well as to the comments submitted by the other reviewers. My comments have been satisfactorily addressed. The revisions to the manuscript language have helped to improve clarity.

Reviewer #3

(Remarks to the Author)

The authors have addressed my comments, and now the conclusions are supported by the data as presented. I would suggest to edit the sentence in abstract: scRNAseq on 69,781 cells from an HGSOc syngeneic mouse model showed significant agreement with the patients' data."

Please consider not using "significant" unless a statistical test has been performed to compare the two groups.

RESPONSE TO REVIEWERS' COMMENTS

Reviewer #1 (Remarks to the Author):

1. The manuscript is quite lengthy and making more concise would increase clarity of manuscript.

Response: We have made changes to the manuscript that we hope improve the clarity and length

2. Validation of the findings in many patient samples by the authors is extremely valuable. However, the study would be greatly enhanced with some spatial context of these populations in human and mouse tumors. The authors should consider including IHC images or performing multiplex IMF on the 7 patients included in original study. Where are these immune cells located? In relationship to each other? To cancer cells? to stroma? Furthermore, while scRNA-seq allowed the authors to identify immune cell populations and gene expression in these immune cells after chemotherapy, other tools are available to define these immune cells including flow cytometry/mass cytometry etc. with greater specificity.

Response: In the submitted manuscript we showed representative images of CD68 and STAB1 IHC in primary and interval debulking samples but did not show examples of FOXP3 staining in the human tumors. Below in Figure R1A and R1B, we show images from omental tumor sections from the 7 patient samples that were used for the scRNA seq analyses, stained by IHC for STAB1 and FOXP3. Figure R1 shows that both STAB1⁺ and FOXP3⁺ cells are widely distributed in both stromal and malignant cell areas. We have now included representative IHC images of STAB1 and FOXP3 staining in paired pre- and post-NACT samples that are quantified in Figures 1G and 1I and Figure 2J (new panels Figure 1H and Figure 2I in the human tumors. We feel that multiplex IMF analysis is beyond the scope of this paper and may not add any more information given the pattern of distribution of the myeloid and FOXP3⁺ cells. We have also added information on the distribution of STAB1 and FOXP3 positive cells in the manuscript and highlighted this in yellow.

Figure R1A. Immunohistochemistry staining of the seven HGSOC human omental samples that were used for scRNAseq (n=7). The staining is done for STAB1. The upper 3 sections are from primary debulking samples (PDS) and the lower 4 sections are from NACT samples. Scale bar at 200µm.

Figure R1B. Immunohistochemistry staining of the seven HGSOc human omental samples that were used for scRNAseq (n=7). The staining is done for FOXP3. The upper 3 sections are from primary debulking samples (PDS) and the lower 4 sections are from NACT samples. Scale bar at 200 μ m.

3. In addition, the figures include a lot of pertinent data but are illegible unless zoom in quite a bit.

Response: We believe that the reviewer was sent a low-resolution version of the figures. Where possible, we have increased the size of the text font in the figure panels.

4. The immune profiling of the mouse models will be extremely valuable as a tool for future researchers so the authors should consider sharing all data raw etc. for others to use to find ideal models to test future therapeutics that activate/exhaust cause infiltration etc. of the immune system

Response: All the raw data will be publicly available on publication under GEO accession numbers. We agree that the data in this paper will be of use to other scientists studying the ovarian cancer tumour microenvironment and future

therapeutics, especially as the HGS2 mouse model is used by others and is freely available via www.cancertools.org.

Reviewer #2 (Remarks to the Author):

1. Are there any flaws in the data analysis, interpretation and conclusions? Do these prohibit publication or require revision?

Lines 568-569 it is written” There is hardly any information on pre-clinical models for relapsed HGSOC.” I am not sure what is meant by this. There are many publications reporting the results of HGSOC models (ex., D.P., Galpin, K.J.C., Rodriguez, G.M. et al. Comparative analysis of syngeneic mouse models of high-grade serous ovarian cancer. *Commun Biol* 6, 1152 (2023). Could the authors please clarify this point?

Response: We thank the reviewer for mentioning the Cook et al paper. However, the comment on lines 568-569 of the original manuscript refers to the experiment in Figure 6F. In this experiment we treated mice bearing 60577 peritoneal tumors that had spontaneously relapsed after a period of remission following successful treatment with chemotherapy, to mimic treatment of relapse in HGSOC patients. Most mouse ovarian cancer experiments in the literature, including the paper referred to by the reviewer, only treated ‘primary’ not relapsed tumors. We have clarified this further in the manuscript and highlighted this point in yellow.

2. The discussion would be further enhanced if mentioned were made of the lack of response of HGSOC to anti-PD1 therapy.

Response: We have added a paragraph to the Discussion showing the lack of response to anti-PD1 in HGSOC and its low expression in this cancer (REF 39).

Reviewer #3 (Remarks to the Author):

1. The main concern is that the analyses of the human samples were done in patient-unpaired samples, introducing an inherent bias towards different tumor characteristics due to patient selection. Therefore, the data cannot be used to support causal claims of NACT effect (e.g. “NACT induced X”). This concerns all of the results that are described on the immune cell subpopulations from the human samples (pages 5-10). The NACT-induced changes in cell phenotypes should be analysed in patient-matched samples, or alternatively the wordings changed throughout the manuscript. The use of patient-unmatched samples also does limit the conclusions that can be made from the study.

Response: We agree with the reviewer that the ideal situation would be to use patient- and site-matched samples. However, this is practically difficult because the time between diagnostic biopsy (pre-NACT sample) and the interval debulking surgery is at least three months and only patients who will have resectable disease after NACT will undergo surgery at this stage. We validated the findings relevant to this paper at the protein level using a larger cohort of 80 patients, 42 of which had matched samples: taken before and after chemotherapy from the same patient. This protein data from the patient-matched samples confirmed our RNA results from the patient-unmatched samples. These data are shown in Figures 1H-1I and 2I-2J.

2. Another major concern is the methodical validity of the scRNAseq analyses. Specifically, some of the filtering cut-offs are not justified, and the differential gene expression analysis presents unusually low cutoff values ($\log_2FC=0.25$). Typically, minimum values of 0.5 or 1 are used to capture the meaningful effect sizes. Similarly, the justifications for p-value cut-offs and multiple hypothesis adjustment are missing. A different set of cutoffs both for the adjusted p-value and \log_2FC are applied in the bulk-RNA seq validation. The statistical analysis needs to be revisited and better explained to demonstrate the validity of the analyses.

Response: To the best of our knowledge, there are no agreed cut-offs in the literature for analysing or interpreting the scRNAseq data. We used the default settings in Seurat4 package functions. The default settings in FindAllMarker and

FindMarkers functions were $\log_{2}FC_{\text{threshold}}=0.25$ and an adjusted p value based on Bonferroni correction. We used this in generating the differentially expressed genes listed in the supplementary tables and for generating volcano plots. However, all the conclusions reported in this paper are mostly reported on average $\log_{2}FC$ of at least 0.5 or above. Another important parameter to consider is the percentage of cells in the cluster expressing a specific gene (pct.1), the higher the pct.1, the more relevant the gene is even if the $\log_{2}FC$ is not as high as another gene has lower pct.1 and very high $\log_{2}FC$.

The values used in bulk RNAseq are different from scRNAseq. This is because in scRNAseq the RNA is from pure populations and several hundreds/thousands of cells and has a better quantification method being a UMI based technology, in contrast to bulkRNAseq which is from whole tissue and is quantified in RPKM. There are also no agreed values for the significant $\log_{2}FC$. For bulk RNAseq, we used $\log_{2}FC$ of 1.

With regards to the p value, there is a consensus that $p < 0.05$ is considered significant and adjusted p value of < 0.05 is used to correct for multiple testing. It is known that in scRNAseq there is an amplification of p value due to the large number of cells (samples) therefore we used even a smaller p value than that used for bulk RNAseq.

We would also point out that we did not base our conclusions on the transcriptomic data solely, but we validated relevant findings at the protein level and conducted series of in vitro experiments to further support the findings and explain mechanisms that we could not study in vivo. Finally, the results in the mouse models support our conclusions.

The detailed analysis of both the scRNA seq and the bulk RNAseq are mentioned in the materials and methods section. In every figure of the paper, the statistical methods of analysis used and the correction for multiple testing are mentioned in the figure legends as well as the materials and methods.

3. In general, the manuscript logic (analyses and figures) and the results are quite hard to follow. The manuscript is partially well but sometimes poorly written as if it was compiled from text written by different persons. The writing should be

harmonised. The manuscript would benefit from clearly focusing on describing the scRNAseq and functional findings of only the most relevant cell populations (e.g. the macrophages and the phagocytosis part is intriguing).

Response: We have revised the manuscript and hope that the clarity has improved.

Specific comments:

1. Fig 1 I (and later similar correlations): PFS correlations should not be made with linear regression - only kaplan-meier analyses should be shown. Anyhow R value of 0.3 barely indicates a significant linear relationship of the variables.

Response: We agree with the reviewer that PFS is not best addressed by linear regression, so we have removed those figures and the relevant parts of the manuscript.

2. Figures 2 and 3 are largely descriptive- focusing on describing the clusters. The text is difficult to follow, the logic of describing the results and the writing should be improved. The main conclusions from this part of the manuscript remain unclear. The authors should rewrite these parts and focus on answering specific scientific questions. It is unclear why the authors focus on STAB1?

Response: As the reviewer highlighted, these data are descriptive, and their main conclusion is that chemotherapy has both favourable and unfavourable effects on anti-tumour immunity. We then focussed the paper on the translational significance of two molecules that we hypothesised could be targeted to improve the response to chemotherapy and studied their actions in vitro and in vivo. We have rephrased the paragraph in the manuscript and hope that the content is clearer and easier to follow.

3. It would be interesting to specify the molecular profile of the patients in which scRNA-seq has been performed, since this cannot be found in either the text or the figures.

Response: We did not conduct bulk RNAseq on the scRNAseq samples used for this paper, therefore it is not possible for us to carry out any molecular classification.

4. The STAB1 PBMC analyses are interesting. What was the rationale to investigate FOXP3 targeted therapy? What was the biological rationale to combine STAB1 and FOXP3 targeted agents?

Response: We chose those two targets because both were induced by chemotherapy and implicated in pro-tumor responses. As this paper describes the pro- as well as anti-tumor effects of chemotherapy, we aimed to target pro-tumor effects to improve chemotherapy responses. STAB1 and FOXP3 were also chosen because they were strongly induced by chemotherapy, and we thought that an optimal response could be achieved if we chose one target associated with the innate and another with adaptive immune responses.

5-From the in vivo work, it is difficult to conclude what are the main findings. Also - the differences in the Kaplan-Meier analyses are not really dramatic, and I wonder how relevant are these differences? Figure 6A-C - it seems that most of the immunotherapy benefit was driven by the FOXP3 - ASO? No significant benefit was gained via adding STAB1? What was the rationale of combining these treatments?

Response: We believe that the results from the in vivo experiments support the hypothesis we generated from the scRNAseq data, IHC and in vitro experiments and confirms the translational significance of our findings and conclusions. Anti-stab1 antibody monotherapy resulted in small but statistically significantly survival benefit (median survival of controls 62 days versus 72 days $P = 0.0004$) but not with FOXP3-ASO monotherapy. However, the addition of FOXP3-ASO or anti-stab1 antibody to chemotherapy resulted in significantly better survival than the standard treatment which is chemotherapy (145 days and 98 days respectively versus 83 days for chemotherapy, $P < 0.0001$ and $P = 0.001$ respectively). The triple combination resulted in even better survival (median survival of 200 days) than any of the dual combinations or the chemotherapy, so part of the effect is due to anti-stab1 antibody treatment, and we confirmed this at the bulk RNAseq level (Figure 7C). Also, we have shown that our suggested combinations resulted in complete cure of some mice and survival over 300

days, and over 50% of these rejected tumours rechallenged. We also found efficacy in treatment of relapsed disease in a third mouse model. As mentioned in the manuscript, disease that has relapsed after chemotherapy is rarely studied in the literature. All the above experiments are the pre-clinical work that would be required before taking this to clinical trial in patients

6. The authors conclude that the anti-stab1 treatment changed the way macrophages process the phagocytosed antigen, potentially strengthening the antitumor immune response. I wonder if there could be a deeper mechanistic characterisation, at the gene expression level, addressing how is the antigen processing different in pre and postNACT settings. In the bulk RNA-seq results from the in vivo experiment tumours they do not go into the differential expression analysis but only to the pathway-level information.

Response: The scRNAseq experiments suggest that chemotherapy upregulated antigen processing and presentation at the pathway and gene level, as shown in Figure 1, but also upregulated scavenger and phagocytic receptors, particularly STAB1 which is a scavenger receptor that directs its cargo for degradation in the lysosomes. We found that chemotherapy also upregulates the pathways of antigen degradation by increasing the acidity of the lysosome at the pathway level and at the gene level by upregulating the lysosomal cathepsins proteases, *CTSB* and *CTSD* (Table S3) which fits in with the STAB1 mechanism of action and outcome.

For the murine bulk RNAseq experiment in Figure 7 we presented pathway results. In Supplementary Table S4, we presented the differential gene expression. We believe that the pathway analyses were more relevant which is why we included them in the main figure.

7-In terms of logic, the discussion needs a lot of work. Instead of discussing the relevance of the findings in a broader context, the authors only discuss the justification of their study and limitations. There are several overstatements in the discussion (for example “*In this paper, we have shown the efficacy of our identified targets in treating relapsed HGSOC”).* None of the 12 ovarian cancer patients with

relapsed disease responded to an Anti-Cleaver-1 antibody treatment

<https://doi.org/10.1016/j.xcrm.2023.101307>

Response: The upregulation of STAB1 on macrophages after NACT has not been reported before and could be relevant to any cancer treated with carboplatin and paclitaxel. We comment on this in the Discussion. We also highlighted that STAB1 and FOXP3 were similarly upregulated after NACT in both the BRCA wildtype and BRCA mutated and HRD positive or negative HGSOE patients which could suggest their potential effect in both groups, unlike the PARP inhibitors which have very limited effect in BRCA-WT and HRD negative patients. We believe this does indicate a wider potential value not only in ovarian but in other cancer using carboplatin and paclitaxel in treatment.

We thank the reviewer for mentioning the anti-cleaver-1 (stabilin-1) Bexmarilimab clinical trial paper. In this paper, 12 patients with HGSOE were given anti-stabilin-1 as monotherapy. No responses were recorded. We believe this paper is relevant to our work for the following reasons:

- a) It showed that Bexmarilimab was a well-tolerated monotherapy which should allow its combination with other drugs similar to those we used in the murine model experiments.
- b) In our murine models, we found marginal increase in survival with anti-stabilin1 antibody monotherapy, however, this was significantly enhanced when combined with chemotherapy or in the triple combination with FOXP3-ASO. Hence, our work suggests the potential of combining chemotherapy with an anti-stabilin-1 antibody.
- c) The molecular findings after anti-stab1 antibody treatment in human samples from different tumors agreed with the molecular effects of anti-stab1 antibody that we reported, such as the increase in CD8 and B cell infiltrate and IFN γ activation, with both the monotherapy and when combined with chemotherapy.

We have now referred to the Bexmarilimab paper in the Discussion (REF 50) and highlighted this addition in yellow.

Minor comments

1. In Fig1A text is hardly visible and it is challenging for the reader to distinguish between clusters. I would suggest increasing font size and changing the colorscale. In Fig. S1E I would add again the legend for the clusters. These comments also apply for Figs 1,2

Response: We increased the font size of the UMAPs in Figures 1-3 but we kept the cluster colours because they are a colour code for the particular clusters in subsequent figure panels.

2. Fig 1D and S1C: annotations of threshold values in the axes are recommended. Colorscale is missing in Fig S1C. Also, the range of the colorscale is not informative in Fig 1D, since color differences are barely noticed. I recommend going for a discrete scale instead of continuous, (>0.05 , $0.05-0.01$, <0.01)

Response: The color scale has been added to Figure S1C. All the P values above the horizontal line are less than $10e-5$. The colorscale is not actually needed to show significance.

3. Cutoff values are too small, fold-change is only 0.25. At least I would recommend a 1 log₂ FC value to account for these differences.

Response: Please refer to Major comments above comment number 2.

4. Lines 128-130: "Compared to PDS samples, we found significant upregulation of macrophage scavenger receptors, e.g., STAB1, MSR1, CD163, phagosome proteolytic cathepsins, e.g., CTSB, CTSL and CTSD and complement in NACT samples" In which figure are these results presented? Some of these genes are not presented in Fig1D.

Response: All the genes that are not in the figures are in supplementary tables (in this case see supplementary Table S3).

5. In Fig 1F, it looks like the expression level of SIGLEC1 was significantly higher in NACT than in PDS?

Response: Yes, it is higher in the NACT group, along with other genes, but we chose to focus on STAB1 because it is expressed by over 75% of the macrophages post-NACT and both murine and human antibodies were available, hence we could study its translational potential. Moreover, although Siglec1 was increased in the human samples post NACT, it was not increased in the HGS2 model, hence we would not have been able to conduct the translational studies.

6. ICGC dataset: Are not all the ICGC dataset tumors PDS? What was the rationale for choosing this dataset?

Response: The only two publicly available HGSOC data sets are TCGA and ICGC and both have PDS samples only, so we choose one of them, ICGC, as an example.

7. Lineage trajectory analyses - what was the scientific question and main result that were obtained from the analyses? It looks like in Fig 1A, for the UMAP the M0 population is found in two separate clusters. Same phenomenon is seen in M2. Also, in the trajectory analysis it seems that one of these clusters is involved in the maturation of M1 and M9 (belonging to the ECM-category), whilst the other seems to be part of lineages comprising APC phenotypes. This is unconventional and needs further clarification.

Response: The aim of this trajectory analysis was to show the potential trajectory connection between the subpopulations of macrophages. The lineage started at monocytes/early macrophages then either went towards the APC macrophages or the ECM macrophages. The method used to perform the trajectory analysis is explained in the materials and methods. We used the previously published Slingshot program (PMID: 29914354) which has been extensively used in other publications.

8. The authors claim that (171): Each patient and both treatment groups had a variable percentage of the dendritic cell clusters with no difference between PDS and NACT. Yet authors show upregulation of AP and phagolysosome-related pathway enrichment in NACT

Response: We are not sure what the reviewer means here. The first part refers to subpopulations and the second part refers to potential function of dendritic cells after NACT. As alluded to in the paper, NACT has a favourable effect on dendritic cells i.e., its upregulation of AP, but also an unfavourable effect as NACT upregulates antigen degradation pathways.

Reviewer #4 (Remarks to the Author):

The authors carried out single-cell RNA sequencing of tumor immune cells from high-grade serous ovarian cancer (HGSOC) samples and carried out validation studies. Please consider the following:

1. Page 5. This reviewer tried to locate Table S1 but could not find it. Additional clinical details on the patients would be important to present, including: 1) what exact chemotherapy drugs did patients receive on NACT? 2) Did any patients receive bevacizumab with their NACT? 3) Did any patients have germline or somatic BRCA mutations or HRD? 4) Did any patients receive maintenance therapy?

Response: We apologise that Table S1 was not included in the original submission. Table S1 is now included. In the materials and methods section, there is a full description of the clinical details of the patients included in this study. To help the reviewer, we have summarised below the main clinical information that is in Table S1 and then have answered in more detail the specific questions from the reviewer.

Summary of clinical information

Twenty-two patients had primary debulking surgery (PDS) and fifty-eight had NACT followed by debulking surgery. All patients were diagnosed with stage IIIC-IV high grade serous ovarian cancer. PDS samples were collected during primary debulking surgery (n=22). NACT samples were collected during interval debulking surgery (n=58), three weeks after the last chemotherapy cycle from patients. Patients received either carboplatin and paclitaxel or carboplatin and paclitaxel plus bevacizumab as NACT, (n=50 and n=8 respectively). Forty-six patients received three-four cycles of NACT, and twelve patients received five-six cycles of NACT. Pre-NACT samples were collected from diagnostic biopsies before the start of treatment (n=42). Clinical data was collected including the age at diagnosis, final histology, response on CT scan after NACT as assessed by RECIST criteria (stable, partial, progressive disease), Chemotherapy Response Score^{24, 25} (CRS1 = majority of the tumour in the omentum is still viable, CRS2= evidence of both tumour apoptosis, fibrosis and viable tumour, CRS3= no viable residual tumor in the omentum), Date of

death or last clinic visit (data was available for 78 patients), date of relapse disease as evidenced by CT scan (data was available for 73 patients in total), chemotherapy regimen in NACT, optimal debulking achieved or not (residual disease) and BRCA/HRD status.

Below we present subgroup analysis:

a) The scRNAseq patient cohort comprised omental metastases samples from three patients receiving primary debulking surgery (PDS) and four patients who had three cycles of neoadjuvant chemotherapy (NACT) followed by interval debulking (IDS). As noted in the original methods, two of these patients received three cycles of carboplatin and paclitaxel (CT) only and two patients received bevacizumab as well as the carboplatin and paclitaxel NACT. All the four NACT patients had a partial response to treatment as assessed by RECIST criteria. To assess if the inclusion of bevacizumab in two of the NACT samples affected our data, we performed differential gene expression analysis between each of the two chemotherapy treatments compared to PDS and found that STAB1 still comes up as a top differentially expressed gene in both groups (Figure R2A and B).

Figure R2. Volcano plot showing differentially expressed genes in macrophages from NACT versus PDS tumors. A) carboplatin and paclitaxel treatment compared to PDS. B) carboplatin, paclitaxel and bevacizumab treatment compared to PDS. Data are from scRNAseq of human HGSOc omental metastasis from 14,672 macrophages. Differential expression was done using default parameters in Seurat package. STAB1 is differentially expressed regardless of the treatment regimen.

b) In the cohort of samples with available FFPE sections, we have also performed subgroup analysis on the NACT cohort (n=58) comparing patients who received carboplatin and paclitaxel only as NACT (n=50) versus those who received bevacizumab as part of their NACT (n=8). We did not find a statistically significant difference in CD68, STAB1 or FOXP3 positive cell number between the two treatment regimens and each of them was significantly higher than the primary debulking surgery (PDS) samples (Figure R3).

c) Additionally, we looked at the STAB1 and FOXP3 levels in tumors from patients with somatic and germline BRCA mutation (BRCAm) tumors, tumors from BRCA wild type (BRCA-WT) patients, homologous recombinant mutation deficiency (HRD). The expression levels of both STAB1 and FOXP3 in BRCA-WT/HRDnegative did not differ compared to the levels in the BRCAm and HRD positive tumors (Figure R4). These data suggest that the treatment we propose in the pre-clinical mouse experiments could potentially benefit both BRCA intact and mutated patients. This was included in the manuscript in Figures S1H and S2F and there is related text in the manuscript.

d) With regards to maintenance therapy, some patients received maintenance treatment either in the form of bevacizumab or PARPi but this was very heterogenous, hospital dependant and were given for variable periods of time and variable reasons to continue/discontinue treatment and was affected by variable confounding factors such as access to hospital follow up during COVID time. So, no conclusion can be made from it.

Figure R4. BRCA status and the levels of STAB1 and FOXP3 in HGSOc omental metastasis. BRCA status was available for 67 of our 80 patients. 40 patients had BRCA wild type (BRCA-WT) and no homologous recombinant deficiency (HRD). 14 patients had either somatic or germline BRCA mutation (BRCAm) and 13 patients had HRD with BRCA-WT. Significance was studied using Mann-Whitney U test.

In summary, the expression of STAB1 and FOXP3 was upregulated by carboplatin/paclitaxel whether bevacizumab was added to NACT or not. Also, the expression was not affected by BRCA/HRD status suggesting the potential benefit in both groups.

2. To compare with human cancer samples, were they selected from patients with BRCA mutation or HRD? Was the histology from the two models used confirmed to be high grade serous?

Response: Regarding BRCA mutation and HRD status of the patients, we have presented these data above and in the main Figures (Figure 1 and 2) and in the manuscript. Tumors from the murine models used in this paper have high-grade serous histology as we previously published (PMID: 31940494).

3. For mouse samples, it would be helpful to see the cell selection criteria prior to scRNA seq.

Response: For the murine samples, after harvesting the fresh omentum and tissue dissociation, we only enriched for live cells to achieve viability above 85% that is compatible with 10X chromium platform. We did not do any further enrichment. In this study, we focused on the data from the murine immune cells only to align with the

human data. This is explained in the materials and methods section. The entire dataset will be publicly available at publication.

4. The authors claim that the cell subpopulations changed after neoadjuvant chemotherapy. However, the cohort used has NACT and PDS samples and these were not matched for each patient. Thus, it is possible that some of the differences might be related to inter-patient heterogeneity rather than chemotherapy. At least some validation in matched samples should be performed.

Response: Please refer to the reply to Reviewer 3 (specific comments section, point number 1).

We validated relevant findings for STAB1 and FOXP3 on patient- and site-matched samples and confirmed our findings (Figure 1H and 2I).

RESPONSE TO REVIEWERS' COMMENTS

Reviewer 1

The authors use 42 patient matched samples to show levels of STAB in tumors but only use n=3 PDS and 4 NAPT patients in sc analysis. Further the authors showed that the overall #s of macrophages, etc. and overall expression level of STAB, etc. was different in pre- vs post-chemo patients (n=42). However, this data did not explore or confirm: "We concluded that NACT increased macrophage infiltration and phagocytic activity leading to 167 increased antigen presentation. However, as NACT also upregulated the macrophage 168 scavenger receptor, STAB1, this could lead to excess antigen degradation and less efficient 169 antigen presentation." Authors did not investigate or validate an increase in STAB expressing populations of macrophages after chemotherapy. As suggested in first review, the authors could address this by including patient-matched and more patients in sc analysis and show UMAPS of both sets and calculate differences M0-M8 populations etc., CD4 cell populations ; the authors could perform co-IMF or multiplex IMF etc to show increase in STAB producing macrophages, FOXP3 expressing CD4 cells; OR the authors could perform flow cytometry on digested tumors as shown in mouse study. The IHC shown and survival analysis are convincing to show that patients with higher tumor levels of STAB...but do not measure infiltration of macrophages and what type of macrophages esp. STAB expressing macrophages.

Response

The data from scRNA seq experiments in unmatched patient samples allowed us to explore potential changes in different immune cell types after NACT that may be used as potential targets. We also confirmed the RNA data in bulk RNAseq samples from a cohort of 16 unmatched samples (supplementary Table 3). It is technically and clinically challenging to get a high-quality sample of sufficient size from a diagnostic biopsy and then, from the same patient, an NACT sample 3-4 months later during surgery. Therefore, we used our scRNA seq data as a guide and went on to validate potential targets at the protein level in matched and unmatched patient samples. As regards defining the macrophage sub-populations that express STAB1, at the RNA level, STAB1 was upregulated in the NACT samples in 8 out of 11 scRNAseq macrophage subpopulations, representing over 75% of the total macrophages. Thus, we concluded that STAB1 upregulation was not linked to a specific subpopulation and could be an important potential target because of its high expression, not only after NACT (75% of macrophages) but also in PDS (around 40% of macrophages). We then studied STAB1 protein expression at protein levels which is possibly more important in translation to pre-clinical or clinical studies.

Specific comments

1. Reviewer 1 states that we did not explore or confirm our conclusions on lines 167-169 i.e. *'We concluded that NACT increased macrophage infiltration and phagocytic activity leading to 167 increased antigen presentation. However, as NACT also upregulated the macrophage 168 scavenger receptor, STAB1, this could lead to excess antigen degradation and less efficient 169 antigen presentation.'*

We showed increased macrophage infiltration after NACT in Figure 1G, but at this point in the paper we had not validated that the correlation between STAB1 expression and antigen degradation and antigen presentation. However, we went on to investigate this in Figure 4. We have therefore amended the text from lines 167-9 to read:

'We concluded that there was an increase in macrophage infiltration and STAB1 expression in the post-NACT samples. We also found significant upregulation of pathways related to scavenger receptors, phagocytosis and antigen presentation but also upregulation of pathways suggestive of increased antigen degradation which could possibly reduce any anti-tumour immune effect of NACT'.

We deleted the last part of the conclusion related to the correlation of STAB1 expression with antigen degradation. We had already mentioned this conclusion after the experiments described in Figure 4 in previous versions of the paper.

2. Reviewer 1 states that *'Authors did not investigate or validate an increase in STAB expressing populations of macrophages after chemotherapy'*. We believe these data are already shown in Figure 1G, 1H and 1J in a cohort of 80 patients, 42 of which have paired patient-matched samples
3. Reviewer 1 has suggested that we *'could address this by including patient-matched and more patients in sc analysis and show UMAPS of both sets and calculate differences M0-M8 populations etc., CD4 cell populations ; the authors could perform co-IMF or multiplex IMF etc to show increase in STAB producing macrophages, FOXP3 expressing CD4 cells; OR the authors could perform flow cytometry on digested tumors as shown in mouse study. The IHC shown and survival analysis are convincing to show that patients with higher tumor levels of STAB...but do not measure infiltration of macrophages and what type of macrophages esp. STAB expressing macrophages.'*

As stated above, we used the patient RNA data, scRNAseq and bulk, as exploratory tools to investigate targets that could improve response to chemotherapy. We validated these targets at the protein level in patient samples and with scRNAseq in a mouse model, further studied potential mechanisms of action *in vitro* and went on to validate hypotheses generated and the translational potential in three different mouse HGSOc models, treating, in two models, established tumours and in the third model, relapsed tumors. We feel that the experiments the Reviewer suggested are outside the scope of this paper and that it would take many months to obtain the samples.

Regarding the comment about STAB1-expressing macrophages, based on scRNAseq data from macrophages in our dataset and in other datasets, tumor-associated macrophage sub-populations exist as a spectrum and not distinct subtypes i.e. the subpopulations express different levels of many of the same markers. It is not technically possible to define these populations at the protein level with currently available tools which are not as sensitive as RNA.

We have conducted some analysis on published scRNAseq and bulk RNAseq datasets and the results support our findings on STAB1 and FOXP3 as explained below:

- 1- In a scRNAseq dataset published by Vazquez-Garcia et al (PMID 36517593) which included treatment naïve patients only, STAB1 was expressed in most macrophage clusters as shown in Figure R1A. Furthermore, similar to our findings, it was expressed in adnexal tumors and in the metastatic sites such as the omentum (Figure R1B and R1C). In the same dataset, Tregs represented around 30% of CD4 T cells in the primary and metastatic sites and are known for their immune-suppressive potential. In our study, we

have shown their prevalence is around 26% and shown the translational benefit of targeting them using a Treg specific siRNA that is available in phase I clinical trials (Figure R2).

2- Further support for our findings comes from a very recently published dataset in Aronson et al, (PMID: 38866963). The dataset included a cohort of 16 HGSOC patients who had interval debulking surgery. Matched Samples were collected before and after NACT for bulk RNAseq. We looked at STAB1 and FOXP3 TPM and showed significantly higher STAB1 and FOXP3 expression post-NACT compared to pre-NACT in this cohort (Figure 3) which agree with our matched cohort of 42 patients and unmatched cohort of 80 patients.

Figure R3. A)STAB1 and B)FOXP3 bulk RNAseq TPM levels from Aronson al et al (PMID: 38866963) of 16 patient matched HGSOC samples taken pre- and post-NACT.

Reviewer 2

As I had previously noted, chemotherapy response of HGSOC to first-line therapy with carboplatin and paclitaxel remains suboptimal, and most patients will go on to have incurable, recurrent disease. The authors present intriguing preclinical findings outlining potential interactions between response to chemotherapy and multiple points of immune regulation. They have shown that both FOXP3 and STAB1 are relevant immune regulators following NACT. They further demonstrate the FOXP3 and STAB1 are potential therapeutic targets in HGSOC, and they present compelling preliminary preclinical data using syngeneic mouse models.

I appreciate the authors response to my comments, as well as to the comments submitted by the other reviewers. My comments have been satisfactorily addressed. The revisions to the manuscript language have helped to improve clarity. The addition of previously published phase 1 clinical trial data helps to contextualize this finding in the clinical research landscape.

It is noted that this manuscript includes the results from multiple experiments using both patient-derived samples and syngeneic mouse models. It can at times be difficult to follow the progression of experiments, as has been noted by other reviewers. However, I do think the adequate information can be discerned from a careful review of the text and figures. I would encourage the authors to be as consistent as possible in their identification of each mode, both in the text and in the attached figures, to improve clarity.

Response

We have tried to improve consistency and clarity in this revised version of the manuscript particularly the abstract, human macrophages and CD4 T cells parts of the results and the

discussion sections. All the revisions made previously were highlighted in yellow and for this second round of revision, revisions are highlighted in blue. We have also made some changes to Figure 1 and 2 and supplementary figure 1 and 2 to increase the clarity of the results. A summary of those additional changes is made at the end of this rebuttal document.

Reviewer #3 (Remarks to the Author):

The authors have addressed most of my concerns. However, question one remains unanswered. While I share the authors' concern on the limitation of access to paired pre and post NACT patient samples, it is critical that the data supports the claims/conclusions made in the manuscript. These, unpaired data cannot be used to support causal claims of **"NACT effect"** (e.g. "NACT induced X"). This starts already in the first sentence of the results (row 105: To study the effect of chemotherapy), as one cannot really study the effect of chemotherapy in unpaired samples. The data would support conclusions that e.g. XX **"was enriched in post NACT"** samples, or that the study investigates the differences in samples before or after chemotherapy. I would ask the authors to tone down here their statements on the results due to this fundamental difference in the research setting and experimental design to avoid misconclusions. This concerns all of the results that are described on the immune cell subpopulations from the human samples. This includes the abstract, results (eg Rows 166-167, 183, 193, 222, 239, 260, 265). Discussion rows 530,

Specific comments.

Comment 1. It would be interesting to specify the molecular profile of the patients in which scRNA-seq has been performed, since this cannot be found in either the text or the figures.

Response: We did not conduct bulk RNAseq on the scRNAseq samples used for this paper, therefore it is not possible for us to carry out any molecular classification.

-> Here I mean BRCA mutation/HRD. Is this information available?

For Specific comments 4-7. In the response letter, the authors don't specify what corrections they made to the manuscript. It would be easier to evaluate how the manuscript has been improved if the changes to the manuscript were annotated in the response letter.

The last paragraph of discussion is nice but would need a language check.

Response:

We apologize for the technical issue that happened. The changes were highlighted in yellow in the manuscript attached as word file. We realised that the highlighted color was not transferred after the automatic conversion of the submitted files into a single merged PDF file.

All the patients are not BRCA mutated or known HRD except one of the PDS patients had an HRD tumour.

We have extensively revised the manuscript and toned-down the conclusions made from the unmatched patient cohorts. This was reflected in the abstract, the manuscript sections on macrophages and Tregs and the discussions. We have edited the last paragraph of the Discussion. We have ensured that the previous yellow and the new blue highlights are clear in the PDF versions.

Reviewer #4 (Remarks to the Author):

Comments have been addressed.

Response: Thank you.

Other changes made to the manuscript and Figures:

We have extensively revised the manuscript and toned-down the conclusions made from the unmatched patient cohorts. Furthermore, we have reanalysed the human IHC data and performed subgroup analyses as explained below.

1- We conducted subgroup analysis for STAB1 expression in the unmatched and matched patient cohort comparing PDS samples and NACT samples in BRCA wild-type and BRCAmutated/HRD patients. This analysis showed that in spite that STAB1 level was not different between BRCA-WT and BRCAm/HRD patients, the increase STAB1 expression after NACT was still true in both groups confirming the potential role for our combination therapies in both groups of patients. This analysis is now in Figures 1G, 1H and 1J and S1G. The patients' data are included in supplementary table 1.

The corresponding figure legends have been changed and highlighted in blue.

The relevant text has been rephrased in the manuscript and highlighted in blue.

2- We conducted the same subgroup analysis for FOXP3 positive cells in the unmatched and matched cohorts of patients comparing PDS and NACT samples in the BRCA-WT and BRCAm/HRD patients. Similar to above, in spite that there was no difference between the FOXP3 levels related to BRCA/HRD status, the increase in FOXP3 levels was still true regardless of the BRCA/HRD status. This analysis is now included in Figures 2H-2I and the S2E. The corresponding figure legends and the relevant part in the manuscript text have been changed and highlighted in blue. The patients' data are included in supplementary table 1.

RESPONSE TO REVIEWERS' COMMENTS

Reviewer #1 (Remarks to the Author):

I appreciate the authors response to my comments, as well as to the comments submitted by the other reviewers. My comments have been satisfactorily addressed. The revisions to the manuscript language have helped to improve clarity.

Reply: Thank you for the reviewer's comments.

Reviewer #3 (Remarks to the Author):

The authors have addressed my comments, and now the conclusions are supported by the data as presented.

I would suggest to edit the sentence in abstract: scRNAseq on 69,781 cells from an HGSOC syngeneic mouse model showed significant agreement with the patients' data." Please consider not using "significant" unless a statistical test has been performed to compare the two groups.

Reply: we agree with the reviewer on removing the word 'significant' and we have removed this now from the abstract to read as:

'ScRNAseq on 69,781 cells from an HGSOC syngeneic mouse model recapitulated the patients' findings'